# OctField: Hierarchical Implicit Functions for 3D Modeling

**Jia-Heng Tang**[*1,2], **Weikai Chen**[*3], **Jie Yang**[1,2],
**Bo Wang**[3], **Songrun Liu**[3], **Bo Yang**[3], **and Lin Gao** (✉)[†1,2]

[1]Beijing Key Laboratory of Mobile Computing and Pervasive Device, Institute of Computing Technology, Chinese Academy of Sciences
[2]University of Chinese Academy of Sciences
[3]Tencent Games Digital Content Technology Center

tangjiaheng19s@ict.ac.cn    chenwk891@gmail.com    yangjie01@ict.ac.cn
{bohawkwang,songrunliu,brandonyang}@tencent.com    gaolin@ict.ac.cn

## Abstract

Recent advances in localized implicit functions have enabled neural implicit representation to be scalable to large scenes. However, the regular subdivision of 3D space employed by these approaches fails to take into account the sparsity of the surface occupancy and the varying granularities of geometric details. As a result, its memory footprint grows cubically with the input volume, leading to a prohibitive computational cost even at a moderately dense decomposition. In this work, we present a learnable hierarchical implicit representation for 3D surfaces, coded OctField, that allows high-precision encoding of intricate surfaces with low memory and computational budget. The key to our approach is an adaptive decomposition of 3D scenes that only distributes local implicit functions around the surface of interest. We achieve this goal by introducing a hierarchical octree structure to adaptively subdivide the 3D space according to the surface occupancy and the richness of part geometry. As octree is discrete and non-differentiable, we further propose a novel hierarchical network that models the subdivision of octree cells as a probabilistic process and recursively encodes and decodes both octree structure and surface geometry in a differentiable manner. We demonstrate the value of OctField for a range of shape modeling and reconstruction tasks, showing superiority over alternative approaches.

## 1 Introduction

Geometric 3D representation has been central to the tasks in computer vision and computer graphics, ranging from high-level applications, such as scene understanding, object recognition and classification, etc, to low-level tasks, including 3D shape reconstruction, interpolation and manipulation. To accommodate with various application scenarios, a universal and effective 3D representation for 3D deep learning should have the following properties: (1) compatibility with arbitrary topologies, (2) capacity of modeling fine geometric details, (3) scalability to intricate shapes, (4) support efficient encoding of shape priors, (5) compact memory footprint, and (6) high computational efficiency.

While explicit 3D representations have been widely used in recent 3D learning approaches, none of these representations can fulfill all the desirable properties. In particular, point cloud and voxel representations struggle to capture the fine-scale shape details – often at the cost of high memory

---

*Contributed equally.

†Corresponding author is Lin Gao (gaolin@ict.ac.cn).

35th Conference on Neural Information Processing Systems (NeurIPS 2021), virtual.

consumption. Mesh-based learning approaches typically rely on deforming a template model, limiting its scalability to handle arbitrary topologies. The advent of neural implicit function [44, 7, 37] have recently brought impressive advances to the state-of-the-art across a range of 3D modeling and reconstruction tasks. However, using only a global function for encoding the entirety of all shapes, the aforementioned methods often suffer from limited reconstruction accuracy and shape generality.

To overcome these limitations, follow-up works have proposed to decompose the 3D space into regular grid [27, 4], or local supporting regions [19], where each subdivided shape is approximated by a locally learned implicit function. The decomposition of scenes simplifies the shape priors that each local network has to learn, leading to higher reconstruction accuracy and efficiency. However, these approaches do not take into account the varying granularities of local geometry, resulting in two major shortcomings. Efficiency-wise, their memory usage grows cubically with the volume of the 3D scenes. Even a moderately dense decomposition could impose severe memory bottleneck. Scalability-wise, the regular gridding has difficulty scaling to high resolutions, limiting its expressiveness when dealing with intricate shapes with small and sharp geometric features (Figure 4).

We observe that most 3D shapes are typically consisting of large smooth regions and small-scale sharp features. In addition, the surface of interest often consumes only a small portion of the entire space, leading to an extremely sparse space occupancy. Based on these observations, we propose a novel 3D representation called *OctField*, that introduces hierarchies to the organization of local implicit functions to achieve better memory efficiency and stronger modeling capacity. As shown in Figure 1, OctField leverages a hierarchical data structure, *Octree*, to adaptively subdivide the 3D space according to the surface occupancy and the richness of geometrical details. In particular, regions enclosing intricate geometries will be further subdivided to allocate more implicit kernels for higher modeling accuracy. In contrast, we stop subdivision for octants containing smooth part geometry as a single implicit kernel would suffice for modeling. Further, we do not allocate

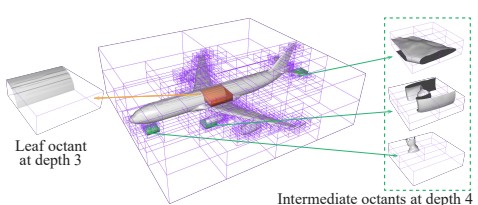

Leaf octant at depth 3

Intermediate octants at depth 4

Figure 1: OctField utilizes an octree structure to achieve a hierarchical implicit representation, where part geometry enclosed by an octant is represented by a local implicit function. OctField achieves an adaptive allocation of modeling capacity according to the richness of surface geometry. In particular, intricate parts such as jet engines, tail-planes and the undercarriage are automatically subdivided to engage more implicit kernels for higher modeling accuracy, while parts with regular shapes on the fuselage is encoded using a coarser-level representation that suffices.

any implicit functions in the unoccupied regions. Hence, OctField could obtain significantly higher representation accuracy with a slightly deeper octree subdivision, as the modeling capacity has been adaptively optimized to accommodate the varying granularity of surface details.

As the octree structure is discrete and non-differentiable, it is non-trivial to directly employ octree in a learning-based framework. We propose a novel hierarchical network that recursively encodes and decodes both octree structure and geometry features in a differentiable manner. Specifically, at the decoder side, we model the subdivision of octree cells as a probabilistic process to make the learning of octree structure differentiable. We employ a classifier to predict whether to subdivide current cell based on its enclosed geometry features. We validate the efficacy of our new representation in a variety of tasks on shape reconstruction and modeling. Experimental results demonstrate the superiority of OctField over the state-of-the-art shape representations in terms of both shape quality and memory efficiency. Our contributions can be summarized as follows:

- A learnable hierarchical implicit representation for 3D learning, named *OctField*, that combines the state-of-the-art hierarchical data structure with local implicit functions.

- A novel hierarchical encoder-decoder network that models the construction of octree as a probabilistic process and is able to learn both discrete octree structure and surface geometry in a differentiable manner.

- We achieve significantly higher surface approximation accuracy with reduced memory cost in the 3D modeling related tasks by using our proposed representation.

## 2  Related Works

**Representations for 3D Shape Learning.** Various 3D representations have been extensively studied in 3D deep learning [28]. These surveys [2, 62] discuss various shape representations comprehensively. As the raw output of 3D scanning devices, point cloud [46, 47, 65, 49, 53] has received much attention in recent years. Despite for its simplicity, generating dense point clouds with high precision remains notoriously difficult. Unlike the other 3D representations, the convolutional network can be directly employed on 3D voxels [36, 11, 20, 59, 60, 61, 64, 23, 10]. Due to the prohibitive computational cost of generating voxels, recent works [55, 57, 56, 52, 32] have introduced octree to the voxel representation to reduce memory cost. The polygon mesh is the another widely used representation in modeling and surface reconstruction. However, current learning-based mesh generation approaches [6, 54, 50, 24, 29, 21, 12, 43] mostly rely on deforming a template mesh, limiting its scalability to shapes with arbitrary topologies. Recent advances in neural implicit functions [44, 37, 7, 27, 14, 63, 8, 45] have significantly improved the surface reconstruction accuracy thanks to its flexibility of handling arbitrary topologies. More recently, [27, 4, 19] have introduced shape decomposition and local implicit functions to further improve the modeling capacity by locally approximating part geometry. [33] introduces implicit moving least-squares (IMLS) surface formulation on discrete point-set to reconstruct high-quality surfaces. However, these methods mostly rely on a regular decomposition and cannot account for sparse surface occupancy and the varying granularities of geometry details, imposing memory bottleneck when dealing with moderately dense subdivision. OpenVDB [42] incorporates $B+$ tree with implicit field to achieve hierarchical modeling. However, the goal of OpenVDB is pursuing extremely fast modeling speed with constant time access in 3D simulation. Hence, the $B+$ tree is non-differentiable and too complex to be incorporated into a learning-based framework. In contrast, OctField is a learnable hierarchical implicit representation that can be differentiably implemented in our hierarchical network. Further, our representation can achieve higher modeling accuracy with even less memory compared to the previous local implicit function approaches [27, 4, 19]. In a concurrent work NGLOD [51], it proposes a similar idea of leveraging level of details to encode local SDFs hierarchically. A corresponding rendering algorithm is proposed to render the neural SDFs in an interactive rate. However, NGLOD cannot learn the hierarchical structure of the underlying octree. In contrast, our hierarchical encoder-decoder network learns the non-differentiable structural information in differentiable manner via modeling it as a probabilistic process. We believe the structural information is crucial for improving the modeling accuracy and future applications(*e.g.* 3D semantic understanding, editing).

**Learning-based Generative Models.** The deep generative models, such as GAN [22] and VAE [30], have shown promising ability of synthesizing realistic images in 2D domain. 3D learning approaches have strived to duplicate the success of 2D generative models into 3D shape generation. 3D-GAN [60] pioneers at applying the GAN technology on the 3D voxels to learn a deep generator that can synthesize various 3D shapes. Generative models on point cloud [1, 15, 67] mainly leverage MLP layers but struggle to generate dense point sets with high fidelity due to the large memory consumption and high computational complexity. Recent works on synthesizing 3D meshes either rely on deforming an initial mesh using graph CNN [54, 58] or assembling surface patches [24, 13] to achieve more flexible structure manipulation. To better model man-made objects composed of hierarchical regular shapes, structural relationship has been considered in [18, 66, 17, 38, 40, 39], where the box-like primitives are used for initial shape to enhance shape regularity. To fully exploit the modeling capacity of implicit surface generator, IM-Net [7] has experimented with both VAE and GAN models to learn stronger shape priors. In a concurrent work, DeepSDF [44] proposes a auto-decoder structure to train latent space and decoder without using a traditional shape encoder. Recently, the local implicit methods [27, 4] combine regular space decomposition with local implicit generators for modeling 3D scenes with fine geometric details. ACORN [35] introduces an adaptive multiscale neural scene representation for 2D and 3D complex scenes, which enables to fit the targets faster and better in an optimized multiscale fashion. In our paper, we propose a hierarchical implicit generative model for 3D modeling. Compared to other methods, our approach can generate high-quality 3D surfaces with intricate geometric details in a memory-efficient manner.

## 3  Method

**Overview.** OctField combines the good ends of both localized implicit representation and the hierarchical data structure. By adaptively allocating local implicit functions according to the surface

occupancy and the richness of geometry, OctField is able to achieve high modeling accuracy with a low memory and computation budget. In particular, we decompose the 3D space into hierarchical local regions using octree structure, where the finest octant encodes the partial shape within its enclosed space using a learned implicit function. Our decomposition protocol not only considers the surface occupancy but also the richness of geometry. As show in Figure 2, the octants that carry an embedded implicit kernel will only be allocated around the surface. Moreover, only the octants containing intricate geometries will be further divided. This ensures an adaptive memory and computation allocation that the richer surface details will be captured with more local implicit functions – hence with higher modeling accuracy. In contrast, the unoccupied regions will not be allocated with any implicit kernels to save the memory and computational budget.

The octree itself is a non-differentiable discrete data structure. We propose a novel differentiable hierarchical encoder-decoder network that learns both the octree structure and the geometry features simultaneously. In particular, we formulate the construction of octree as a probabilistic process where the probability of subdividing an octant is predicted by a MLP layer. This makes it possible to learn discrete octree structure in a fully differentiable manner. In addition, we train our network in a VAE manner such that the trained latent space and decoder can be used for a variety of downstream applications including shape reconstruction, generation, interpolation, single-view reconstruction, etc. We provide detailed description of the OctField representation and the proposed network in Section 3.1 and 3.2 respectively.

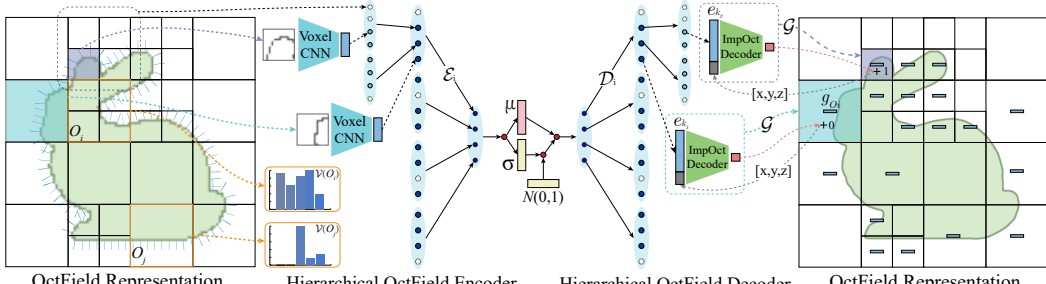

OctField Representation     Hierarchical OctField Encoder     Hierarchical OctField Decoder     OctField Representation

Figure 2: 2D illustration of our hierarchical OctField network. We propose a novel recursive encoder-decoder structure and train the network in a VAE manner. We use the voxel 3D CNN to encode the octants' geometry, and recursively aggregate the structure and geometry features using a hierarchy of local encoder $\{\mathcal{E}_i\}$. The decoding is implemented using a hierarchy of local decoders $\{\mathcal{D}_i\}$ with a mirrored structure with respect to the encoder. Both the structure and geometry information are recursively decoded and the local surfaces are recovered using the implicit octant decoder within each octant.

### 3.1 OctField Representation

**Octree Construction.** To build an octree for the input model, we first uniformly scale the 3D shape into an axis-aligned bounding box and then recursively subdivide the bounding region into child octants in a breadth-first order. The octant to be subdivided has to satisfy two requirements simultaneously: (1) the octant encloses the surface of interest; and (2) its enclosed geometry needs to have sufficient complexity that is worth subdividing. We use the normal variance of the surface as an indicator of its geometric complexity. Specifically, we formulate the normal variation of a surface patch $S$ as follows:

$$\mathcal{V}(S) = \mathbb{E}_i(\mathcal{V}(\{\mathbf{n}_x^i\}) + \mathcal{V}(\{\mathbf{n}_y^i\}) + \mathcal{V}(\{\mathbf{n}_z^i\})) \tag{1}$$

where the $\mathbf{n}_x^i, \mathbf{n}_y^i, \mathbf{n}_z^i$ are the $x, y, z$-component of the normal vector $\mathbf{n}^i$ at the $i$-th sampling point on the surface; $\{\mathbf{n}_x^i\}$ denotes the collection of $\mathbf{n}_x^i$; $\mathcal{V}(\cdot)$ calculates the variations of the input while $\mathbb{E}_i(\cdot)$ returns the expectation. In our experiments, we perform regular sampling on the surface where the sampling points are pre-computed. We repeat the decomposition until the pre-defined depth $d$ is reached or $\mathcal{V}(S)$ is smaller than a pre-set threshold $\tau$. We set $\tau = 0.1$ throughout our experiments.

**Local Implicit Representation.** The implicit function associated with each octant is designed to model only part of the entire shape. This enables more training samples and eases the training as most 3D shapes share similar geometry at smaller scales. At each octant, the enclosed surface is

continuously decoded from the local latent code. However, as the finest octant may have different sizes, when querying for the value of the local implicit function, we normalize the input world coordinate $\boldsymbol{x}$ against the center of the octant $\boldsymbol{x}_i$. Formally, we encode the surface occupancy as: $f(\boldsymbol{c}_i, \boldsymbol{x}) = \mathcal{D}_{\theta_d}(\boldsymbol{c}_i, \mathcal{N}(\boldsymbol{x} - \boldsymbol{x_i}))$, where $\mathcal{D}_{\theta_d}$ is the learned implicit decoder with trainable parameter $\theta_d$, $\boldsymbol{c}_i$ is the local latent code and $\mathcal{N}(\cdot)$ normalizes the input coordinate into the range of $[-1, 1]$ according to the bounding box of the octant. To prevent the discontinuities across the octant boundaries, we propose to enlarge each octant such that it overlaps with its neighboring octant at the same level. In our implementation, we let each octant has $50\%$ overlap along the axis direction with its neighbors. When the implicit value at the overlapping regions is queried, we perform tri-linear interpolation over all the octants that intersect with this query position.

## 3.2 Hierarchical OctField Network

To enable a differentiable framework for learning the octree structure and its encoded geometry, we propose a novel hierarchical encoder-decoder network that organizes local encoders and decoders in a recursive manner. We embed both the octree structure information and the geometry feature into the latent code of each octant. As shown in right part of Figure 2, the latent code $\boldsymbol{e}_i = (\boldsymbol{g}_i, \boldsymbol{\alpha}_i, \boldsymbol{\beta}_i)$ for octant $O_i$ is a concatenation of three parts: (1) a geometry feature $\boldsymbol{g}_i$ that encodes the local 3D shape; (2) a binary occupancy indicator $\boldsymbol{\alpha}_i$ that indicates whether the octant encloses any 3D surface; and (3) a binary geometry subdivision indicator $\boldsymbol{\beta}_i$ that denotes whether the enclosed geometry is intricate enough that needs further subdivision. We will show in the following subsections that how this configuration of latent vector guides the recursive decoding and encoding in our network. Note that, unlike the prior tree structure-based generative models [66, 38, 31], our approach does not require a manually labeled part hierarchy, e.g. the PartNet [41] dataset, for training, and can generate the hierarchical structure automatically using our octree construction algorithm.

### 3.2.1 Hierarchical Encoder

As shown in Figure 2, the encoder $\mathbf{E}$ of our network is composed of a hierarchy of local encoders $\{\mathcal{E}_i\}$ that encodes local geometry feature and octree structure into the latent code. While our framework supports general geometry encoders, we employ a 3D voxel CNN $\mathcal{V}$ for extracting geometry features due to its simplicity of implementation. After constructing the octree for input model, we voxelize the surface enclosed in each octant in a resolution of $32^3$.

The encoding process starts from the octants at the finest level in a bottom-up manner. For each octant $O_i$, we first compute its binary indicators $(\boldsymbol{\alpha}_i, \boldsymbol{\beta}_i)$ according to its enclosed geometry. In particular, $\boldsymbol{\alpha}_i$ is set to 1 if there exist surfaces inside $O_i$ and 0 if otherwise; $\boldsymbol{\beta}_i$ is set to 1 if $O_i$'s enclosed geometry (if $\boldsymbol{\alpha}_i = 1$) satisfies the subdivision criteria as detailed in Section 3.1 and 0 if otherwise. We then extract $O_i$'s geometry feature $\boldsymbol{g}_i$ by passing its enclosed voxelized geometry $G_i$ to the voxel CNN $\mathcal{V}$. When proceeding to a higher level, our network will

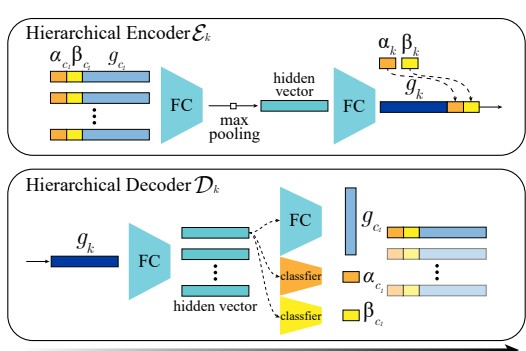

Figure 3: The architecture of hierarchical encoder $\mathcal{E}_k$ and decoder $\mathcal{D}_k$. $\mathcal{E}_k$ gathers the structure $(\boldsymbol{\alpha}_{c_j}, \boldsymbol{\beta}_{c_j})$ and geometry $\boldsymbol{g}_{c_j}$ feature of child octants to its parent octant $k$ by a MLP, max-pooling operation, and another MLP, where $c_j \in \boldsymbol{C}_k$. $\mathcal{D}_k$ decodes the parent octant feature $\boldsymbol{g}_k$ to features $\{\boldsymbol{g}_{c_j}\}$ and two indicators $\boldsymbol{\alpha}_{c_j}, \boldsymbol{\beta}_{c_j}$ of its child octants by two MLPs and classifiers. Two indicators infer the probability of surface occupancy and the necessity of further subdivision, respectively.

aggregate the children's latent features to its parent octant. In particular, for a parent octant $O_k$, we denote the octant features of its children as $\{\boldsymbol{e}_{c_j} = (\boldsymbol{g}_{c_j}, \boldsymbol{\alpha}_{c_j}, \boldsymbol{\beta}_{c_j}) \,|\, c_j \in \boldsymbol{C}_k\}$, where $\boldsymbol{C}_i$ represents the child octants of $O_k$. Its encoder $\mathcal{E}_k$ then aggregates the latent features of $O_k$'s child octants into $O_k$'s geometry feature $\boldsymbol{g}_k$:

$$\boldsymbol{g}_k = \mathcal{E}_k\left(\boldsymbol{e}_{c_0}, \boldsymbol{e}_{c_1}, \cdots, \boldsymbol{e}_{c_7}\right). \tag{2}$$

We then obtain $O_k$'s latent feature by concatenating $\boldsymbol{g}_k$ with $O_k$'s structure features $(\boldsymbol{\alpha}_k, \boldsymbol{\beta}_k)$. We perform the recursive feature encoding and aggregation until the root node has been processed. Specifically, the encoder $\mathcal{E}_i$ consists of a multi-layer perceptron (MLP), one max pooling layer and another MLP for output. At the end of encoder, we leverage the VAE re-parameterization technique

to encourage the distribution of the latent space to fit a normal distribution. Note that all the local encoders $\mathcal{E}_i$ share its parameters to leverage the similarity of local geometries and to reduce the network parameters.

### 3.2.2 Hierarchical Decoder

The hierarchical decoder $\mathbf{D}$ aims to decode the octree structure and local octant codes from the input global feature. It consists of a hierarchy of local decoders $\{\mathcal{D}_i\}$ with a mirrored structure with respect to the encoder $\mathbf{E}$. On the contrary to $\mathbf{E}$, the decoding process starts from the root node and recursively decodes the latent code of its child octants in a top-down manner. Specifically, for a parent octant $O_k$ with geometry feature $\boldsymbol{g}_k$, we decode the geometry features of its child octants using the decoder $\mathcal{D}_k$:

$$(\boldsymbol{e}_{c_0}, \boldsymbol{e}_{c_1}, \cdots, \boldsymbol{e}_{c_7}) = \mathcal{D}_k (\boldsymbol{g}_k), \tag{3}$$

where $c_j \in \boldsymbol{C}_k$ denotes the child octant of $O_k$ and $\boldsymbol{e}_{c_j} = (\boldsymbol{g}_{c_j}, \boldsymbol{\alpha}_{c_j}, \boldsymbol{\beta}_{c_j})$ stands for the geometric feature and two indicators of the child octant $O_{c_j}$. The two indicators provide the probability of whether the child octants need to be decoded or subdivided. Note that we decode all the 8 child octants at one time.

In particular, $\mathcal{D}_k$ consists of two MLPs and two classifiers (see Figure 3). We first decode $\boldsymbol{g}_k$ into hidden vectors $\boldsymbol{v}_{c_j}$ for all 8 child octants by a MLP. To decode the structure information, we apply two classifiers $\mathcal{I}_g$ and $\mathcal{I}_h$ to infer the probability of surface occupancy and the necessity of further subdivision, respectively. For child octant $O_{c_j}$, we feed its hidden vector $\boldsymbol{v}_{c_j}$ into $\mathcal{I}_g$ and $\mathcal{I}_h$, and calculate $\boldsymbol{\alpha}_{c_j} = \mathcal{I}_g(\boldsymbol{v}_{c_j})$ and $\boldsymbol{\beta}_{c_j} = \mathcal{I}_h(\boldsymbol{v}_{c_j})$. For predicting the $\boldsymbol{g}_{c_j}$, we apply the other MLP on $\boldsymbol{v}_{c_j}$. If $\boldsymbol{\alpha}_{c_j} \leq 0.5$, it indicates that $O_{c_j}$ does not contain any geometry and will not be further processed. If $\boldsymbol{\alpha}_{c_j} > 0.5$, it means that $O_{c_j}$ is occupied by the surface and we will further check the value of $\boldsymbol{\beta}_{c_j}$. If $\boldsymbol{\beta}_{c_j} \leq 0.5$, we will not further subdivide the octant and will infer its enclosed surface using the implicit octant decoder $\mathcal{G}$ and the geometric feature $\boldsymbol{g}_{c_j}$. If $\boldsymbol{\beta}_{c_j} > 0.5$, we will proceed to subdivide the octant by predicting the latent features of its child octants with the same procedure. We repeat this process until no octants need to be subdivided.

**Implicit Octant Decoder.** We use a local implicit decoder $\mathcal{G}$ to reconstruct the 3D surface within the octant. For octant $O_i$, we feed its latent geometry feature $\boldsymbol{g}_i$ and the 3D query location $\boldsymbol{x}$ to the implicit decoder $\mathcal{G}$ for occupancy prediction. We train $\mathcal{G}$ with binary cross entropy loss on the point samples. The training loss for octant $O_i$ is: $\mathcal{L}_{geo} = \frac{\sum_{j \in \mathcal{P}} \mathcal{L}_c(\mathcal{G}(\boldsymbol{g}_i, \boldsymbol{x}_j), \mathcal{F}(\boldsymbol{x}_j)) \cdot w_j}{\sum_{j \in \mathcal{P}} w_j}$, where $\mathcal{F}(\cdot)$ returns the ground-truth label (inside/outside) for input point, $\mathcal{L}_c(\cdot, \cdot)$ is the binary cross entropy loss, $\mathcal{P}$ denotes the set of sampling points, $w_j$ describes the inverse of sampling density near $\boldsymbol{x}_j$ for compensating the density change as proposed in [7]. Note that $\mathcal{G}$ is pre-trained on all the local shape crops to encode stronger shape prior.

In order to obtain stronger supervision, we strive to recover the local geometry of all the octants that are occupied by the surface regardless if it belongs to the finest level. Hence, the total loss for training our hierarchical encoder-decoder network is formulate as follows:

$$\mathcal{L}_{total} = \mathbb{E}_{O_i \in \boldsymbol{O}}[\lambda \mathcal{L}_{geo} + \mathcal{L}_h + \mathcal{L}_k + \beta \mathcal{L}_{KL}], \tag{4}$$

where $\mathcal{L}_h$ and $\mathcal{L}_k$ denote the binary cross entropy loss of classifying whether the octant contains geometry and needs to be subdivided, respectively, $\mathcal{L}_{KL}$ is the KL divergence loss, and $\mathbb{E}[\cdot]$ returns the expected value over the set of all octants $\boldsymbol{O}$ that enclose surface geometry. We set $\lambda = 10.0, \beta = 0.01$ throughout our experiments.

## 4  Experiments

In the section, we will first introduce our data preparation process and then evaluate our approach in a variety of applications, including shape reconstruction, shape generation and interpolation, scene reconstruction and shape completion. We also provide ablation study, more comparisons and implementation details in the supplemental materials.

### 4.1  Data Preparation

Our network is trained and evaluated on the five biggest and commonly used object categories in the ShapeNet dataset [5]: chair, table, airplane, car, and sofa. For fair comparison, we use the

officially released training and testing data splits. All the shapes are normalized to fit a unit sphere and converted into watertight meshes [26] for computing ground-truth signed distance field. We first build the octree for each model according to the protocol defined in Section 3.1. To account for the sparse occupancy of surfaces, we apply importance sampling to sample more points near the surface and exponentially decrease the point density as the distance to surface increases. We sample 10000 points in total for each octant, and calculate its corresponding signed distance to surface. To deploy 3D CNN, we adaptively voxelize the part shape in each octant to ensure it maintains the $32^3$ resolution regardless the size of the host octant. We use the voxelization code provided by [25].

**Metrics.** We follow the commonly used reconstruction metrics: Chamfer distance (CD) [3] and Earth Mover's Distance (EMD) [48] for quantitative evaluation and comparison with prior methods.

## 4.2 Shape Reconstruction

In this section, we evaluate the performance on shape reconstruction and compare with the following state-of-the-art approaches that are closely related with our method: IM-Net [7], OccNet [37], Local Implicit Grid (LIG) [27], ConvOccNet [45], Adaptive O-CNN (AOCNN) [57], and OGN [52]. In particular, IM-Net and OccNet use a global implicit function to depict the the entirety of 3D shape while LIG decomposes the input shape into regular grid and leverages local implicit kernels to approximate part geometry. AOCNN and OGN also utilizes the octree structure. However, instead of using local implicit functions, AOCNN uses a single plane to approximate the local geometry enclosed in each octant. OGN predicts

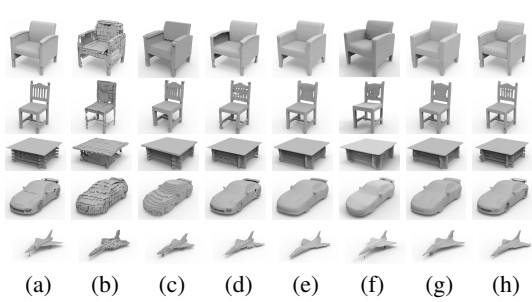

(a)  (b)  (c)  (d)  (e)  (f)  (g)  (h)

Figure 4: Shape reconstruction comparison with the baseline methods ((a) Input, (b) AOCNN [57], (c) OGN [52], (d) LIG [27], (e) OccNet [37], (f) ConvONet [45], (g) IM-Net [7], and (h) Ours).

occupancy in octant without further local geometric feature. We show the visual comparisons in Figure 4. While IM-Net and OccNet are capable of reconstructing the global shape of the object, they fail to reconstruct detailed structures. LIG can recover some of the fine-scale geometries but has difficulty in modeling sharp and thin structures as shown in the second row. Since AOCNN only uses a primitive plane to approximate local geometry, it suffers from the discrepancy between adjacent octants and cannot recover complex local structures due to its limited approximation capability. In comparison, our approach achieves the best performance in all categories and is able to faithfully reconstruct intricate geometry details, such as the slats of chair backs, the hollowed bases of the tables and and the wheels of the cars.

We report the result of quantitative comparisons in Table 2, 1. Our approach outperforms the alternative approaches over all categories and achieves the best mean accuracy in terms of CD, EMD, mIoU, and F1 score metrics. In particular, our reconstruction accuracy is significantly higher than IM-Net, OccNet and AOCNN over all categories.

| Method | IM-Net | OccNet | LIG | Ours |
|--------|--------|--------|------|------|
| mIoU↑ | 79.9 8 | 71.36 | 86.28 | **87.96** |
| F1↑ | 0.83 | 0.70 | 0.93 | **0.94** |

Table 1: Quantitative evaluation on shape reconstruction, we report mIoU and F1 score.

**Computational Cost.** In Table 3, we compare the computational cost with the local implicit approach using regular decomposition – LIG [27]. We show the consumption of local implicit cells/octants used for surface modeling and the computation memory with an increasing decomposition level. Thanks to our adaptive subdivision, our approach consumes significantly less local kernels compared to LIG to achieve similar or even better modeling accuracy. This leap becomes more prominent with

|  | level | 1 | 2 | 3 | 4 |
|--|-------|---|---|---|---|
| Number of cells | LIG | 8 | 64 | 512 | 4096 |
|  | Ours | 8 | 30 | 200 | 1000 |
| Memory (GB) | LIG | 0.1 | 0.6 | 5 | 40 |
|  | Ours | 0.2 | 1.2 | 4.8 | 23 |

Table 3: Comparisons of computational cost with LIG [27]. We show the consumption of the local cells and memory with respect to different levels of decomposition.

| Dataset | IM-Net | | Occ-Net | | LIG | | AOCNN | | ConvOccNet | | OGN | | Ours | |
|---|---|---|---|---|---|---|---|---|---|---|---|---|---|---|
| | CD | EMD | CD | EMD | CD | EMD | CD | EMD | CD | EMD | CD | EMD | CD | EMD |
| Plane | 4.21 | 3.39 | 5.62 | 3.46 | 2.50 | 2.57 | 6.90 | 4.26 | 3.03 | 3.82 | 7.43 | 4.61 | **2.29** | **2.47** |
| Car | 15.14 | 4.46 | 13.54 | 4.93 | 5.46 | 4.08 | 16.61 | 5.63 | 10.04 | 5.66 | 16.24 | 6.23 | **4.84** | **2.79** |
| Chair | 6.99 | 3.77 | 7.87 | 4.16 | 2.37 | 2.18 | 10.80 | 6.76 | 3.98 | 3.19 | 10.77 | 5.65 | **2.19** | **2.13** |
| Table | 8.03 | 3.16 | 7.47 | 3.34 | 2.81 | 2.27 | 9.15 | 4.78 | 3.83 | 3.04 | 9.03 | 3.88 | **2.53** | **1.71** |
| Sofa | 7.95 | 2.51 | 8.6 | 2.81 | 3.23 | 2.06 | 9.39 | 3.49 | 4.03 | 2.85 | 8.79 | 4.32 | **3.02** | **1.84** |
| Mean | 8.46 | 3.45 | 8.62 | 3.74 | 3.27 | 2.63 | 10.57 | 4.98 | 4.98 | 3.712 | 10.45 | 4.94 | **2.97** | **2.19** |

Table 2: Quantitative evaluation on shape reconstruction. In this table, we report the CD ($\times 10^{-4}$) and EMD ($\times 10^{-2}$) scores (smaller is better) on five categories. OctField can achieve the best performance on average score and each category by comparing to six baselines (IM-Net [7], OccNet [37], Local Implicit Grids [27], Adaptive O-CNN [57], ConvOccNet [45] and OGN [52]).

the increasing level of decomposition. As OctField requires additional memory to maintain octree structure, at low decomposition level, our memory cost is slightly higher than LIG. However, with a finer subdivision, our memory consumption drops significantly and becomes much lower than that of LIG. It indicates an increasing advantage of our method in modeling intricate geometry in higher resolution.

### 4.3 Shape Generation & Interpolation

As we train our network in a VAE manner, our model is able to generate diversified 3D shapes by feeding our pre-trained decoder with random noise vectors sampled from a normal distribution. Our network learns a smooth latent space that captures the continuous shape structures and geometry variations. To generate novel 3D shapes, we randomly sample a latent vector in our learned latent space, and decode it into

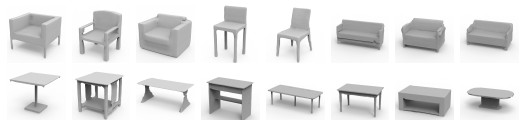

Figure 5: Shape Generation. We show the results generated by randomly sampling the latent codes in the latent space.

shape space by extracting its zero-isosurface using MarchingCubes [34]. In Figure 5, we show the generated results on chair and table categories respectively. Despite the random sampling, our approach is still able to synthesize high-quality 3D shapes with complex structure and fine geometric details, e.g. the second and the fourth table in the second row.

Another approach to synthesize new shapes is to interpolate between the given shapes in the latent space. For two input shapes, we interpolate their latent codes linearly and feed the obtained latent vectors to the pre-trained decoder for shape interpolation. Figure 6 shows the interpolated results on the chair and table categories. Our approach can achieve smooth and continuous interpolation even between two highly diversified objects with distinct structures. In addition, the sharp geometry features, e.g. the six-square-grid base of the table in the

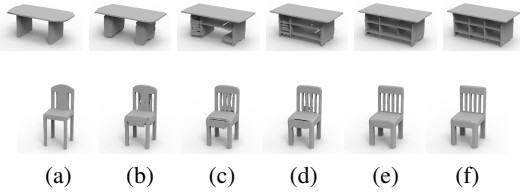

| (a) | (b) | (c) | (d) | (e) | (f) |

Figure 6: Shape Interpolation. The figure shows two interpolated results in two categories: table and chair. (a) is source shape, (f) is target shape.

first row, can be well maintained during the interpolation. This indicates that our network is capable of learning a smooth manifold to generate novel shapes in high fidelity.

### 4.4 Scene Reconstruction

Compared with a single object, our representation is more advantageous when dealing with large scenes. Our representation can obtain better reconstruction details while saving computational overhead. In this section, we illustrate the superiority of OctField on large scene dataset 3D-Front [16]. Further, we compare it with local implicit approach using regular decomposition – LIG [27], convolutional occupancy network [45], NGLOD [51] and ACORN [35] quantitatively and qualitatively. In Figure 7, we present two camera views in a large scene from 3D-Front [16] dataset.

From the visualization results, we can observe that our results is capable of capturing more fine-grained geometric and structure details compared to LIG. The other two methods that introduce hierarchical structure also perform well. It is worth mentioning that high-quality visualized results can be generated by rendering

| Method | $CD_{\times 10^{-4}}$ | $EMD_{\times 10^{-2}}$ |
|--------|------|------|
| LIG | 7.1 | 36.1 |
| ConvOccNet | 10.5 | 22.3 |
| NGLOD | 11.3 | 33.1 |
| ACORN | 7.7 | 24.2 |
| Ours | **6.4** | **21.1** |

Table 4: Quantitative evaluation on scene reconstruction.

directly from the SDF of NGLOD[51]. However, extracting the mesh from the implicit field could cause loss of reconstruction accuracy. Table 4 shows quantitative comparison results. Our approach outperforms the alternative approaches over the large scene models and achieves the best performance in terms of CD and EMD metrics.

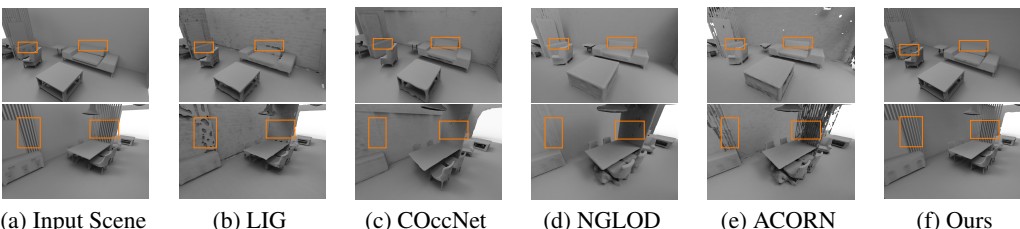

| (a) Input Scene | (b) LIG | (c) COccNet | (d) NGLOD | (e) ACORN | (f) Ours |

Figure 7: Scene Reconstruction and Comparison. In this figure, some large scene reconstruction and comparison with Local Implicit Grid [27], convolutional occupancy network (Conv OccNet) [45], NGLOD [51] and ACORN [35] are presented. We show that our method can provide more accurate reconstruction of geometric and structural details of large scenes. The experiment are performed on 3D-Front [16].

## 4.5 Shape Completion

We evaluate our method in the task of shape completion. Furthermore, We compare with the IF-Net [9] and and demonstrate that our method achieves more robust completion performance with less artifacts in Figure 8. Specifically, we first voxelize the partial point cloud and then map it to the latent space of OctField representation via a 3DCNN encoder. The retrieved latent code is fed to our hierarchical decoder for reconstructing the octree structure, as well as the geometric surface. For IF-Net, we also adopt same voxelization and map the partial voxels to the complete shape by its shape completion model.

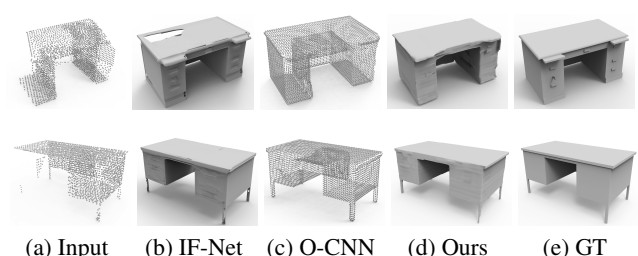

| (a) Input | (b) IF-Net | (c) O-CNN | (d) Ours | (e) GT |

Figure 8: Shape Completion and Comparison with IF-Net [9] and O-CNN [56]. Our method is able to recover complete and faithful 3D shapes only from partial point clouds.

Shape completion results (see Figure 8 and Table 5) on the table category show that our method achieves more robust completion performance with less artifacts. Compared to another octree-based method [56], our method predicts the complete mesh of partial input rather than dense point cloud.

| Method | IF-Net | O-CNN | Ours |
|--------|--------|-------|------|
| $CD(\times 10^{-4})$ | 4.9 | 12.1 | 4.4 |

Table 5: Quantitative evaluation on shape completion.

## 4.6 Shape Editing

With the proposed differential octree generative model, our framework enables some potential applications, such as part editing that modifies or replaces only part of the target geometry. In order

to realize parting edit, we re-parameterize the latent code of the partial local shape, introducing the local VAE to modify and replace the local geometric shape. In Figure 9, we show the results of the part editing of our method comparing with a naive method of directly blending two implicit fields from the source and target shapes. Our approach can generate a smooth transition even between two distinct structures while the naive blending method cannot guarantee a continuous connection for local shape editing.

## 5  Conclusions and Discussions

We have proposed a novel hierarchical implicit representation for 3D surfaces, coded OctField. OctField takes advantages of the sparse voxel octree representation to adaptively generate local supporting regions around the surface of interest. By associating a local implicit function with each octant cell, OctField is able to model large-scale shape with fine-level details using compact storage. To accommodate the non-differentiable nature of octree, we further propose a novel hierarchical network that models the octree construction as a probabilistic process and recursively encodes and decodes both structural and geometry information in a differentiable manner. The experimental results have shown superior performance of OctField over

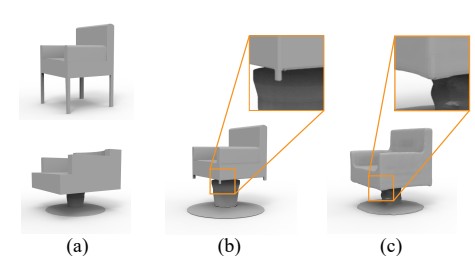

Figure 9: Part editing. The figure shows two edited chairs using our network. ((a) source, (b) blending the SDF directly, (c) our result)

the alternative approaches in a variety of tasks related to shape modeling, reconstruction, and editing. In the future, we would like to incorporate semantic meaning into the organization of octree to encode structural information and enable flexible editing of part-level geometry. In addition, it is also an interesting avenue to explore adaptive length of local latent code such that local implicit functions with higher modeling capacity are only dealing with geometries with more intricate details.

## 6  Broader Impact

The proposed OctField can serve as a fundamental representation of 3D geometry and thus can have a positive impact in a broad range of research fields, including computer vision, computer graphics, and human-computer interaction, etc. Specifically, due to the cost-effective nature of our representation, our method can reduce the economic cost of 3D environment acquisition from raw scanning, while maintaining a high-fidelity modeling performance. This could benefit a number of real-world applications, including modeling large-scale 3D scenes, compressing and transmitting high-quality 3D models for telecommunication and telepresence. Our generative model can also be used for low-cost 3D shape generation without the need of performing actual 3D scanning and post processing, which are expensive and time-consuming. However, at the same time, special care must be taken not to violate the privacy and security of the private scene owners during the process of data collection for our model training.

**Acknowledgments.** This work was supported by CCF-Tencent Open Fund, the National Natural Science Foundation of China (No. 61872440 and No. 62061136007), the Beijing Municipal Natural Science Foundation (No. L182016), the Royal Society Newton Advanced Fellowship (No. NAF\R2\192151) and the Youth Innovation Promotion Association CAS.

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
