# OctField: Hierarchical Implicit Functions for 3D Modeling
## – *Supplemental Material* –

**Jia-Heng Tang**[*1,2], **Weikai Chen**[*3], **Jie Yang**[1,2],
**Bo Wang**[3], **Songrun Liu**[3], **Bo Yang**[3], **and Lin Gao** (✉)[†1,2]

[1]Beijing Key Laboratory of Mobile Computing and Pervasive Device, Institute of Computing
Technology, Chinese Academy of Sciences
[2]University of Chinese Academy of Sciences
[3]Tencent Games Digital Content Technology Center

tangjiaheng19s@ict.ac.cn    chenwk891@gmail.com    yangjie01@ict.ac.cn
{bohawkwang,songrunliu,brandonyang}@tencent.com    gaolin@ict.ac.cn

## Overview

In this supplemental material, we provide more details on network architecture and more visualization results, including shape reconstruction/comparison, shape Generation, and shape Interpolations. Furthermore, some results on scene reconstruction and comparison with Local Implicit Grid [3] are presented to demonstrate our superiority on large data representation thanks to the hierarchical tree structure of our proposed OctField representation.

All sections are listed as follows:

- Section 1 provides the details of network architecture and training.
- Section 2, Section 3 and Section 4 provide more visualization results on a number of 3D modeling tasks, including shape reconstruction, generation and interpolation.
- Section 5 conducts four ablation studies, including with or without overlapping of adjacent octants, the training strategy, the distinction of latent codes and the subdivision parameter $\tau$.

## 1 Network Structure and Training Details

Our network consists of a hierarchical encoder $\mathbf{E}$ and a hierarchical decoder $\mathbf{D}$. Hierarchical encoder $\mathbf{E}$ includes a 3D voxel encoder $\mathcal{V}$ and a hierarchy of local aggregators $\{\mathcal{E}_i\}$, which extract the shape feature from 3D voxels and aggregate features from child octants to its parents octants along the given octree structure, respectively. Note that all the hierarchical aggregators $\{\mathcal{E}_i\}$ share the same network parameters.

Hierarchical decoder $\mathbf{D}$ includes an implicit octant decoder and some hierarchical local decoders $\{\mathcal{D}_i\}$, which maps the decoded shape code and a sample point $(x, y, z)$ to the local geometry and the local latent feature, respectively. Note that all of the hierarchical local decoders $\{\mathcal{D}_i\}$ share the same network parameters and the mesh surface is recovered by MarchingCubes [5] with 32768 sample points within each octant.

The brief architecture of hierarchical encoder and decoder are shown in Figure. 3 of our main paper. In Figure 1, we present the detailed architecture of encoder, which includes two FCs (Single Layer

---

[*]Contributed equally.
[†]Corresponding author is Lin Gao (gaolin@ict.ac.cn).

35th Conference on Neural Information Processing Systems (NeurIPS 2021), virtual.

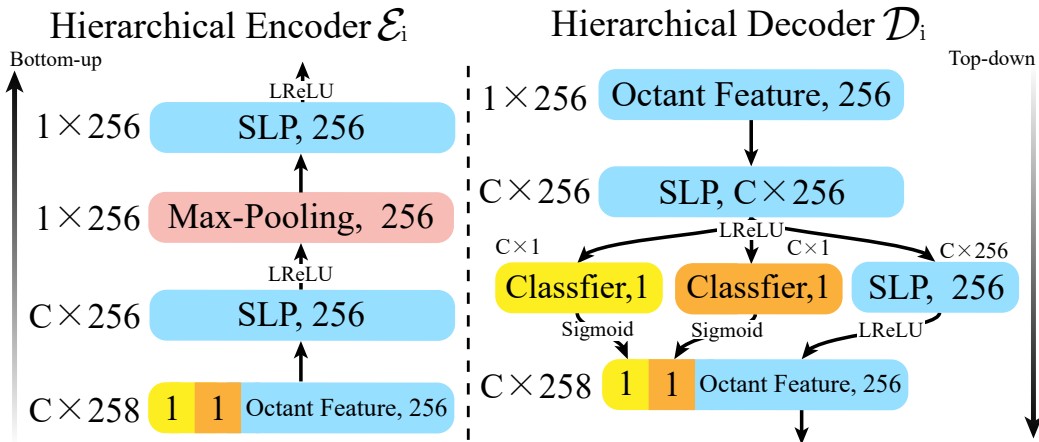

Figure 1: The detailed architecture of hierarchical encoder $\mathcal{E}_i$ and decoder $\mathcal{D}_i$. $\mathcal{E}_i$ gathers the structure and geometry feature of child octants to its parent octant in a bottom-up manner by a FC, max-pooling operation, and another FC. $\mathcal{D}_i$ decodes the feature of parent octant to features and two indicators of its child octants in a top-down manner by two FCs and classifiers. Two indicators infer the probability of surface occupancy and the necessity of further subdivision, respectively. All the feature dimension of octants and latent space are 256. The LeakyReLU is applied as the activation, but for two classifiers, we use the *Sigmoid* function to predict the probability of two indicators.

| Layer | Kernel size | Stride | Padding | Activation function | $(C_{out}, D, H, W)$ |
|---|---|---|---|---|---|
| Input voxels | – | – | – | – | (1,32,32,32) |
| Conv3d | (4,4,4) | (1,1,1) | (1,1,1) | IN, LeakyReLU($\alpha = 0.02$) | (32,31,31,31) |
| Conv3d | (4,4,4) | (2,2,2) | (1,1,1) | IN, LeakyReLU($\alpha = 0.02$) | (64,15,15,15) |
| Conv3d | (4,4,4) | (2,2,2) | (1,1,1) | IN, LeakyReLU($\alpha = 0.02$) | (128,7,7,7) |
| Conv3d | (4,4,4) | (2,2,2) | (1,1,1) | IN, LeakyReLU($\alpha = 0.02$) | (256,3,3,3) |
| Conv3d | (3,3,3) | – | (0,0,0) | – | (64,1,1,1) |

Table 1: The network architecture of 3D Voxel Encoder. IN means the instance normalization 3D operator [9]. The activation is LeakyReLU, and the negative slope is 0.02.

Perceptron) and a max-pooling operation. The decoder includes two FCs and two classifiers for geometric and structure feature prediction of child octants. All the feature dimension of octants and latent space are 256. For the leaf octants, we map the feature (64) into 256-d feature vector by a FC with a LeakyReLU ($\alpha$=0.02) activation for aggregating until to root octant. Before inputting the feature vector into implicit octant decoder, we apply another FC without activation to map the 256-d feature vector into 64-d shape code for predicting the probability of occupancy.

We provide two tables (Table 1 and Table 2) to detail the network structures of 3D voxel encoder and implicit decoder, respectively.

During network training, we first pre-train the 3D voxel encoder and implicit decoder. Then the hierarchical network can be optimized under the fixed implicit octant decoder and 3D voxel feature extractor. All the parameters of network are initialized by Xavier initialization [2]. The network is trained with batch size 32 and consumes approximately 2~3 GB GPU memory. For implicit octant decoder, we randomly sample 4096 points from overall sampled points (10000) in the forward pass.

## 1.1 Implementation Details

We implement our network in PyTorch [7]. We use the Adam solver [4] for parameter optimization. All the parameters of network are initialized by Xavier initialization [2]. During training, we first train the local geometry decoder over all the octants. Then the hierarchical network is optimized in an end-to-end manner with the pre-trained implicit octant decoder and the voxel feature extractor. For implicit octant decoder, we randomly sample 4096 points from overall sampled points (10000)

| Layer | $H_{in}$ | Activation function | $H_{out}$ |
|---|---|---|---|
| shape code + (x, y, z) | (64 + 3) | – | (67) |
| fully-connected | (67) | LeakyReLU($\alpha = 0.02$) | (256) |
| fully-connected | (256) | LeakyReLU($\alpha = 0.02$) | (256) |
| fully-connected | (256) | LeakyReLU($\alpha = 0.02$) | (256) |
| fully-connected | (256) | LeakyReLU($\alpha = 0.02$) | (128) |
| fully-connected | (128) | LeakyReLU($\alpha = 0.02$) | (64) |
| fully-connected | (64) | LeakyReLU($\alpha = 0.02$) | (32) |
| fully-connected | (32) | Sigmoid | (1) |

Table 2: The network architecture of Implicit Octant Decoder. The activation is LeakyReLU, and the negative slope is 0.02. In the last layer, we use *Sigmoid* activation to predict the probability of occupancy of sampled point within each octant.

in the forward pass. Note that all of the hierarchical local decoders $\{\mathcal{D}_i\}$ share the same network parameters and the mesh surface is recovered by MarchingCubes [5] with $32^3$ sample points within each octant. we set the batch size as 32 and use learning rate 0.001 which decays every 100 steps with ratio 0.9, until the network converges about 2000 steps. In the post-processing, meshes far away from the body of the shape has been removed and laplacian smoothing was performed. Empirically, our network converges within 1 days, which is trained with a single GeForce RTX 2080Ti and an Intel i9-9900K CPU.

## 2 Shape Reconstruction

In this section, we show more shape reconstruction and comparison results with four alternative methods (IM-Net [1], Adaptive O-CNN [10], Local Implicit Grid [3], Occupancy Network [6], Convolutional Occupancy Network [8]) qualitatively in figure 2. From the visualized results, we can observe that our approach is capable of capturing more accurate geometric details compared to the other methods, such as the table leg, chair back (5-th row), and the wheel of car.

## 3 Shape Generation

In this section, we present more visualized results on shape generation for the table and chair categories. Our network learns a smooth latent space that captures the continuous shape structures and geometry variations. To generate novel 3D shapes, we randomly sample a latent vector in our learned latent space, and decode it into shape space by extracting its zero-isosurface using MarchingCubes [5]. In Figure 3, we show more randomly generated shapes by our generative network in table and chair category. Our results shows the diversity of the generated shapes in terms of structure variations and geometric details.

## 4 Shape Interpolation

In this section, we present some interpolated results (see Figure 4) on the table and chair categories. For two given shapes, we map them into the latent space, and then interpolate them linearly. All of the shapes are decoded from our latent space. In all figures, we can observe that our interpolated sequence is capable of performing smooth transition between highly diversified structures and geometries.

## 5 Ablation Study

In the section, we perform two ablations studies to demonstrate the necessity and effectiveness of some key algorithmic components. First, we evaluate the effectiveness of introducing overlap between the neighboring octants for resolving the surface discrepancy between octants. Then, we compare the separate training and end-to-end training for the implicit octant decoder and hierarchical octant network.

## 5.1 With *v.s.* without overlapping.

Since our approach seeks to locally approximate the part geometry using implicit functions, the boundary regions between adjacent octants may present discontinuity if it is not properly handled. Unlike the regular decomposition [3] where the local cells have the identical sizes, the adjacent octants in our method may have different scales due to different levels of subvidision. Hence, we propose to introduce adaptive overlapping mechanism that only extent octant at the same level with a ratio of 50% along three axis At inference, the implicit values at overlapping region will be obtained by interpolating the predictions over the intersecting octants. In Table 3 and Figure 5, we compare the qualitative and quantitative results with and without this strategy. We show that our overlapping mechanism successfully prevents disconnections and achieve much smoother and more accurate reconstructions.

| Metric | Overlap (Ol) | | Training Strategy | |
|---|---|---|---|---|
| | With Ol | Without Ol | Separate | End-to-End |
| CD↓ $\times 10^{-4}$ | **2.29** | 4.67 | **2.29** | 14.37 |
| EMD↓ $\times 10^{-2}$ | **2.46** | 2.56 | **2.47** | 6.02 |

Table 3: In this table, we show two ablations experiments (overlap, training strategy) on airplane category. The quantitative results demonstrate that our result have better performance on two metrics (CD & EMD) by overlapping octant in separate training scheme.

## 5.2 Separate training *v.s.* End-to-end training.

We experiment with two training strategies. One is to train the hierarchical encoder-decoder structure along with the implicit octant decoder in an end-to-end manner, coded as "End-to-end" training. The other is to pre-train the implicit octant decoder using part geometries and then fine-tune it with the rest of the hierarchical network, which is referred to "Separate training". In Table 3 and Figure 5, we evaluate the performance of two training strategies. We observe that the separate training outperforms end-to-end training both quantitatively and qualitatively. It is primarily due to that the pre-training of local octant decoder have obtained strong shape priors by learning from a large amount of part shapes, which can facilitate the subsequent fine tuning in an end-to-end manner. In contrast, the end-to-end training method hampers a randomly initialized local geometry decoder from learning local shape prior more efficiently.

## 5.3 Classification

By incorporating and in a hierarchical manner, which explicitly encodes the structural information and the hierarchy of the octree and the geometry, it leads to more discriminative features in the latent space, that can significantly boost the performance of the subsequent applications, e.g shape reconstruction. We verify this point by conducting a control experiment that replaces the proposed hierarchical encoder with a PointNet++ encoder for encoding the point cloud. On the task of shape reconstruction, experimental results show that our encoder has significantly outperformed Pointnet++ in terms of chamfer distance (our hierarchical encoder: 2.19 vs. Pointnet++ encoder: 13.54) on the ShapeNet chair category. Meanwhile, we visualized the result of binary classification using t-SNE in Figure 6.

## 5.4 The subdivision threshold $\tau$

Although the division depends on not only the value of $\tau$ but also a maximum depth, it is necessary to select a suitable value to reduce the division without significantly affecting the accuracy. If the value $\tau$ is set to 1, then the root node will not be subdivided and our method will indeed become equivalent with an OccNet. However, if the value $\tau$ is set to 0, the octree will be divided to the maximum depth. Furthermore, the ablation on $\tau$ is listed in Table 4. Finally, we selected $\tau = 0.1$ to achieve a balance between accuracy and efficiency.

| $\tau$ | 0 | 0.1 | 0.5 | 0.8 |
|---|---|---|---|---|
| CD($\times 10^{-4}$) | 2.96 | 2.97 | 4.87 | 8.42 |

Table 4: Ablation on $\tau$ ($\tau = 0, 0.1, 0.5.0.8$). Comparing to always subdividing to the maximum depth($\tau = 0$), similar reconstruction accuracy can be achieved by choosing $\tau = 0$.

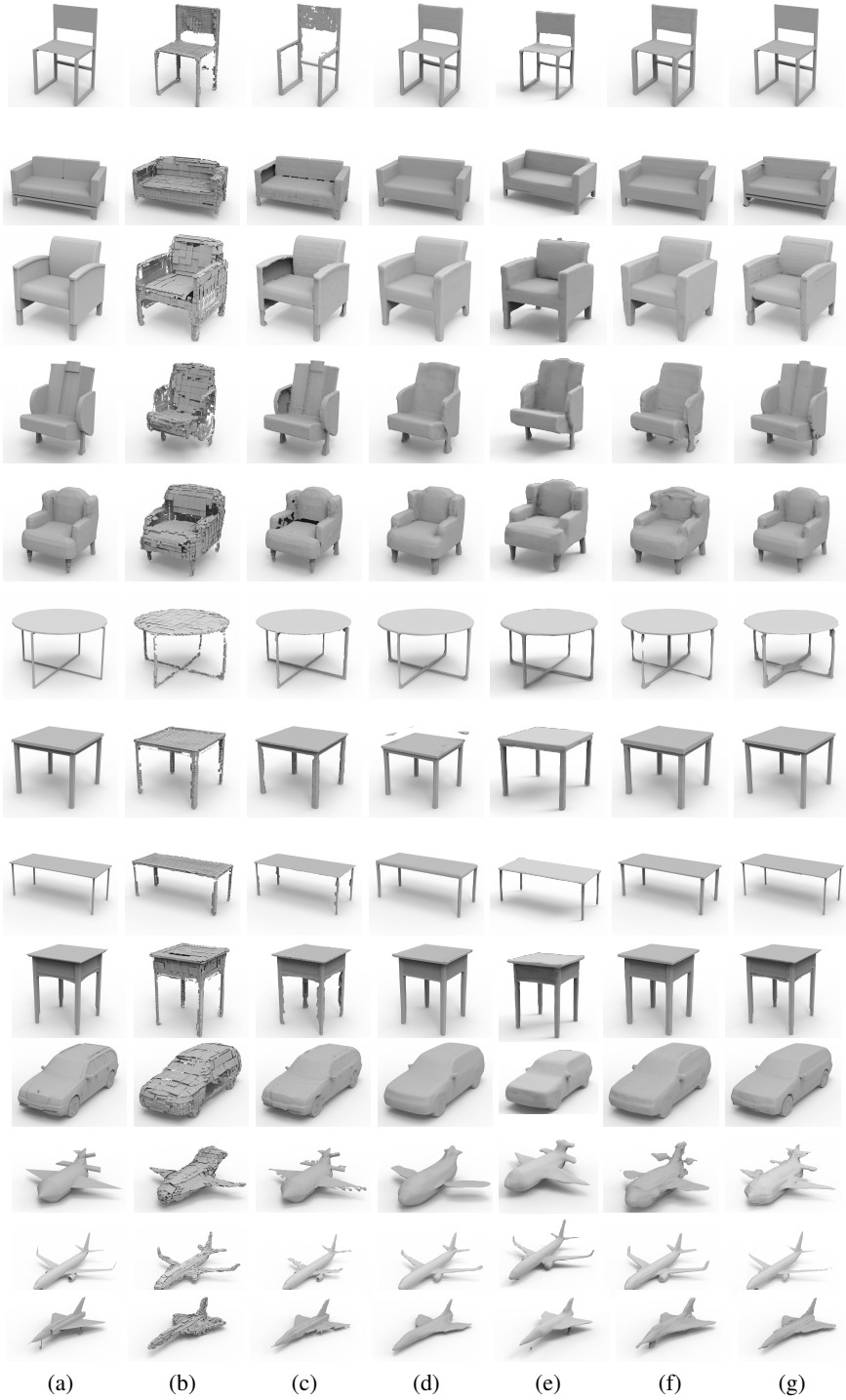

| (a) | (b) | (c) | (d) | (e) | (f) | (g) |

Figure 2: Shape reconstruction comparison with the baseline methods ((a) Input, (b) AOCNN [10], (c) LIG [3], (d) OccNet [6], (e) ConvONet [8], (f) IM-Net [1], and (g) Ours).

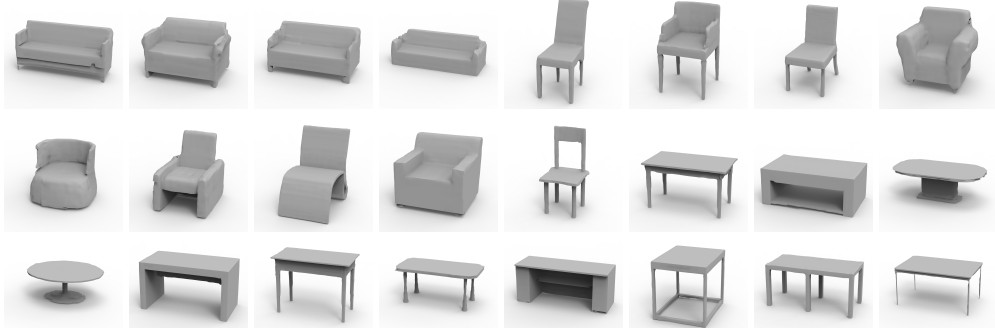

Figure 3: Shape Generation. The figure shows some generated results in table and chair categories. For each generated shape, we randomly sample a latent vector in latent space and decode it by our learned hierarchical decoder. Our results shows our method is capable of capturing the diversity of shape on structure changes and geometry details in learned latent space.

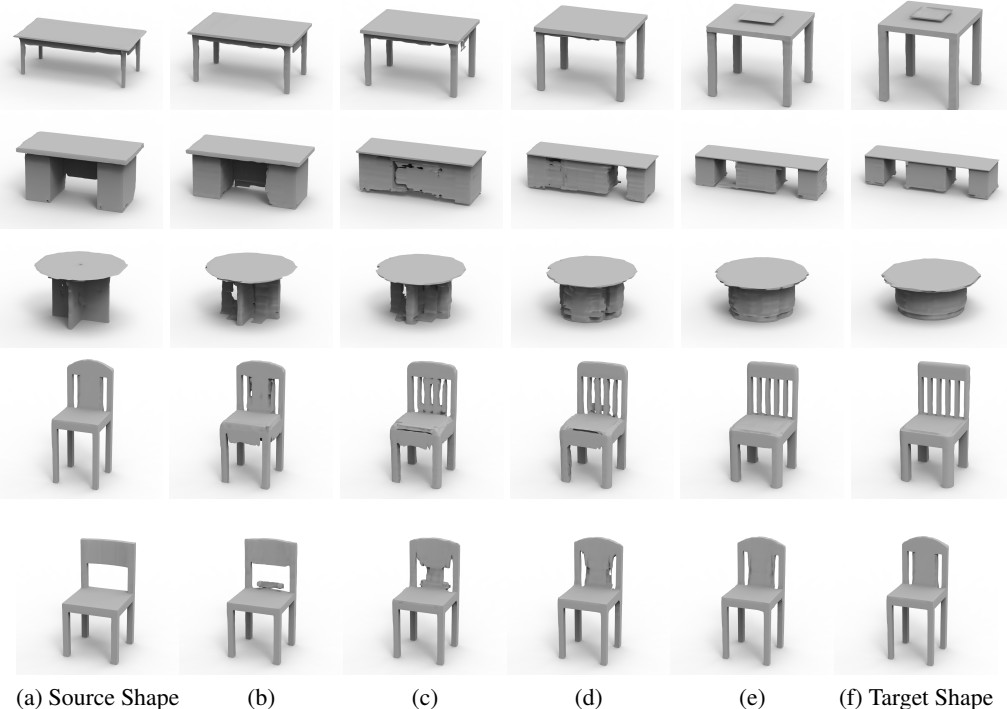

(a) Source Shape    (b)    (c)    (d)    (e)    (f) Target Shape

Figure 4: Shape Interpolation. The figure shows three interpolated results in two categories: table and chair. We show that our approach can achieve smooth and continuous change of shape even when interpolating highly diversified objects with distinct structures. The first and last columns are the decoded source (a) and target shapes (f). The other shape (b),(c),(d),(e) are the interpolated shapes between source and target shapes.

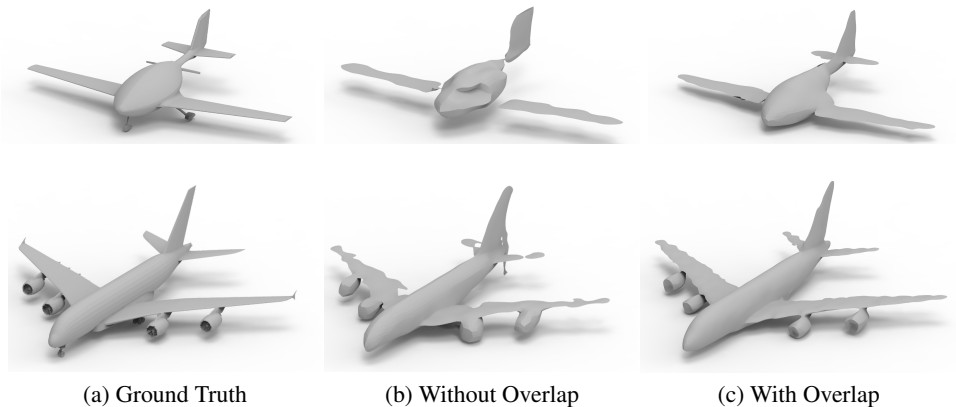

|  (a) Ground Truth | (b) Without Overlap | (c) With Overlap |

Figure 5: The ablation study on training strategy and octant overlap. The first row displays the evaluation on separate training and joint training. The second row displays the evaluation on with *v.s.* without overlap. The figure demonstrates that the separate training and the overlapping strategy achieve better performance compared to its counterpart approach in terms of qualitative results.

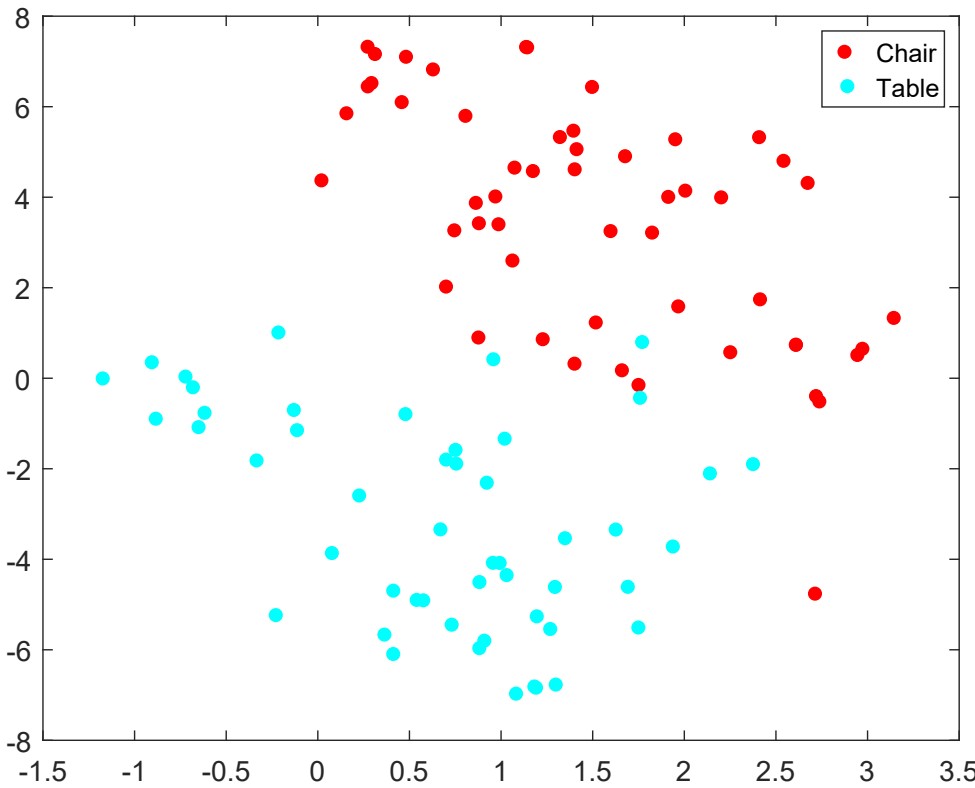

Figure 6: The t-SNE visualization shows that our method can generate discriminative features between chairs and tables.