# OpenReview forum: "OctField: Hierarchical Implicit Functions for 3D Modeling"
_NeurIPS.cc/2021/Conference — NeurIPS 2021 Poster_

### Official Review · Reviewer_Nodw · 2021-07-07

**Rating:** 6
**Confidence:** 4

**Summary:**

The paper proposes a hierarchical implicit grid representation for modeling implicit shape-spaces. While concurrent research has focused either on hierarchical representations in overfitting mode, or on fixed grids in shape-space mode, the paper proposes a technique that simultaneously satisfies these two objectives. The method is tested on a few classes from the synthetic SDF'ied shapenet against a few methods, and has seemingly good reconstruction performance.

**Ethical Concerns:**

None.

**Limitations And Societal Impact:**

No, and it's fine.


**Main Review:**

The ability of modeling adaptive/learnable hierarchical representations is a very exciting research direction, and this paper is one the few that has been looking at it.
However, the paper makes the mistake to focus the entire analysis on reconstruction accuracy, rather than showing the capabilities to interpolate between trees with drastically different topologies (and all the ablations / explanations that justify why a VAE model make this possible).
This, together with the low'ish quality of the writing (see the many, too many, questions and suggestions below), is what makes me rate the paper as a weak reject.
**I might raise my score to a weak accept, as a way to be friendly to the fact that NGLOD partially scooped the work, as far as my questions are properly answered.** (RAISED)


1. Q: what happens if you disable the VAE loss? I was amazed not to see this ablation, nor a visualization of how the topology of the representation changes during interpolation tasks.
1. Q: it is very difficult to understand the quality of a latent space from screenshots... do you have a video? Further, in this video a side-by-side comparison to the IM-Net VAE shape-space (to understand the differential gain caused by the hierarchical model) is essential?
1. Q: can you justify why you only use 5 ShapeNet classes? (this is usually a red flag that results were cherry picked, unless you provide a very strong reasoning)
1. Q: while CD is commonly used, EMD is rarely seen, while mIOU (for solids) and F1 are much more common. Can you justify why were they not reported?
1. Q: I am quite concerned about Table2, where I was expecting a **significantly** bigger advantage in terms of memory vs. LIG (main competitor with a fixed grid). You argue the gains are more evident as you consider deeper trees, but provide no rationale on why subdivisions at 5,6,7 levels were not reported? (note L7 is the standard Poisson reconstruction subdiv level). Further, it would have been good to see the CD scores in this analysis as well... Why were they omitted?
1. Q: Discontinuities solved "a-la LIG" is not super elegant, increases memory usage (overlap) and causes visible artefacts (top-right of Fig.4). Why is the NSVF/NGLOD solution of interpolate latent codes not feasible in your setting?
1. Q: how can one (me) be sure that the generative model has not just retrieved training samples? (show closest model in the training set w.r.t. CD from the generated result to certify this)

Finally, while ACORN / NGLOD are contemporary, I was quite disappointed not to see some qualitative comparisons (would have not affected the rating).


## Further questions (Q) and suggestions to the authors

1. Q: what were the improvements that CVPR/TVCG asked, and that you applied? I was expecting to see this in the supplementary
1. Q: why did you not discuss "Neural Sparse Voxel Fields" from NeurIPS'20? While this is radiance rather than SDF based, the underlying hierarchical structure is very similar
1. There is a disturbing amount of grammatical mistakes that a careful proofread by a native speaker would be able to fix. e.g. at lines L{88,89,149,193,207,214,228,235,247,278}, and this list is not exhaustive
1. please sort citations in `\cite` environments! [3,18,26] vs. your [26,3,18], it makes it faster to look up what you are referring to
1. I can see why you discuss generative modeling in the related works after reading the technical/results sections, but it was not clear at all when reading the introduction. The reasoning is that you are proposing a hierarchical shape-space, while ACORD/NGLOD are mostly overfitting. As such, I would **strongly** recommend you move the discussion of concurrent work to the intro, and mention there the focus on generative modeling? Currently the structure of intro/related sections confused me a little.
1. Q: I am quite confused about the notation surrounding (1)... why do you need two aggregation operation? If $\mathcal{V}$ is variance, then it contains an expectation, but then you also have $\mathbb{E}$. Yet, the metric is the one of a patch, so I would expect only one expectation in this expression...?
1. Q: Why is (1) more meaningful that a more classical the divergence of the gradients within the patch (which is then related to curvature)? This is what Poisson surface reconstruction uses, coupled with a density estimator (like what you do for weighting the computation of the loss)
1. Fig.2, you'd mention that the encoders are shared, also, if they are... why bother with giving them indices?
1. Q: I think you made a mistake in preparing the figure, as the VAE constraints should affect both mean and variance, while you only show it on the variance branch? why?
1. L199: undefined $\mathbf{E}$ in Fig.2?
1. L204 and L204: overloaded CNN $\mathcal{V}$ and variance operator $\mathcal{V}$?
1. Q: why is the VAE prior only applied to the **global** latent code? You have a hierarchy, how do you justify not having it at every level?
1. L243: a bit more precision than `classifiers` would be good... is it just a linear layer with a two-class softmax?
1. L260: the importance of pre-training is enormous (supplementary), and should be quickly mentioned in the main text.
1. L273: summarizing the set of experiments in the supplementary is a good practice?
1. Q: in Sec 4.2, I would recommend to call this is auto-encoding (reconstructing shape collections) rather than reconstruction (reconstructing a single shape/overfitting).
1. L304: comparisons in "c"?
1. L316: you are referencing the wrong table
1. Q: can you justify the weird failure case of LIG in Fig.4? I would expect LIG to just have slightly smoother results than your method. If there is a mistake in how the comparison is performed, the metric in Table.1 (for LIG) could be completely different.
1. Table2: very unclear whether this is an amortized results (average) or whether it's on a single object/scene?
1. Q: Sec 4.4: while I am pretty sure in 4.2 you intend auto-encoding, I am similarly pretty sure in this section you are referring to overfitting. If so, why not compare to methods like SIREN?
1. Q: Sec 4.5: why are you only showing qualitative results? If no quantitative is possible, a clear rationale should be provided!

**Time Spent Reviewing:**

3

---

> ### Author Response · Authors · 2021-08-10
> **Response to Reviewer Nodw**
>
> We thank the reviewer for the insightful comments and suggestion! While we address the main concerns below, we will also fix other minor problems on grammar, notations, figures, equations, tables, and citations in the revision.
>
> ### Main Questions
> > What happens if you disable the VAE loss? ... nor a visualization of how the topology of the representation changes during interpolation tasks.
>
> We have performed an ablation study of disabling the VAE loss. If the VAE loss is disabled, the reconstruction quality is slightly better, but the network loses the ability of novel shape generation and smooth shape interpolation. We do provide a visualization of the topology change during the interpolation task in Figure 5. Please refer to it for more details.
>
> > It is very difficult to understand the quality of a latent space from screenshots...? Further, in this video a side-by-side comparison to the IM-Net VAE shape-space ...is essential?
>
> We will provide the video to compare with IM-Net side-by-side. For the shape interpolation, the intermediately interpolated shapes generated by IM-Net are changed globally while our method is able to interpolate the partial shapes within some local octants.
>
> > Can you justify why you only use 5 ShapeNet classes? ...
>
> The five categories are not cherry-picked. We use 5 representative classes following the prominent previous works on implicit shape reconstruction - DeepSDF [1] and IM-Net [2], which also only evaluate on 5 categories. In particular, these five classes are the largest 5 categories with the most complex structure and geometric details. We believe by evaluating on these categories can provide a fair evaluation of our model. If requested, we are happy to evaluate our method on the other categories.
>
> [1] DeepSDF: Learning Continuous Signed Distance Functions for Shape Representation, CVPR'19
>
> [2] Learning Implicit Fields for Generative Shape Modeling, CVPR'19
>
> > While CD is commonly used, EMD is rarely seen, while mIOU and F1 are much more common.
>
> We provide the quantitative results measured in mIOU and F1 as follows and will add these results to the revision.
>
> ||IM-Net|OccNet|LIG|Ours
> |:-:|:-:|:-:|:-:|:-:|
> |mIOU|79.98|71.36|86.28|**87.96**
> |F1|0.83|0.70|0.93|**0.94**
>
> > ... why subdivisions at 5,6,7 levels were not reported? ... it would have been good to see the CD scores in this analysis as well... Why were they omitted?
>
> In the following table, we show the comparison between LIG and our method on scene reconstruction in terms of memory cost and reconstruction accuracy. Different from Poisson reconstruction, we find that our method can achieve sufficient reconstruction accuracy at level 3 due to the strong modeling capacity of an implicit decoder. Further increasing the depth will not bring us significantly improved accuracy. This enables us to strike a good balance between accuracy and memory cost. In contrast, LIG still needs higher depth for more accurate reconstruction while the memory cost increases much faster than ours. We are not able to run LIG in level 7 due to the overwhelming memory that LIG requires.
>
> |level|2|3|4|5|6|7|
> |:-:|:-:|:-:|:-:|:-:|:-:|:-:|
> |LIG(Memory, GB)|0.6|5|40|320|2500|20000|
> |Ours(Memory, GB)|1.2|4.8|23|100|400|1500|
> |LIG(CD, $\times10^{-4}$)|5.21|3.27|3.23|3.15|3.02|-|
> |Ours(CD, $\times10^{-4}$)|4.82|2.97|2.95|2.93|2.92|2.92|
>
>
>
> > Discontinuities solved "a-la LIG" is not super elegant, ... NSVF/NGLOD solution of interpolate latent codes not feasible in your setting?
>
> In the early research period, we were inspired by LIG and found the overlapping solution works well in resolving discontinuity in our framework. However, the recent solution of interpolating latent code is also feasible in our method. We will experiment with it and provide an ablation study in the revision.
>
> > How can one (me) be sure that the generative model has not just retrieved training samples?
>
> We have evaluated the generative ability of our network by randomly sampling our latent space. We find the generated shapes are unseen in the training set (Fig. 5). Furthermore, we also observe that the interpolated results are different from the source and target shape in Fig. 6, which demonstrates that our network is able to generate novel shapes instead of just retrieving training samples.
>
> > Finally, while ACORN/NGLOD are contemporary, I was quite disappointed not to see some qualitative comparisons (would have not affected the rating).
>
> We conducuted additional experiments on scene reconstruction with NGLOD (ACORN hasn't released its code yet). The reconstruction accuracy of NGLOD is CD($\times10^{-4}$)/EMD($\times10^{-2}$)=11.3/33.1 while ours=6.4/21. In addition, unlike NGLOD and ACORN, which only focus on reconstructing the input geometry with high accuracy, our framework is a **generative model** that is capable of 1) reconstructing input geometry with high fidelity; and 2) generating novel hierarchical octree and its corresponding geometry. With the proposed hierarchical encoder-decoder network, our method is able to learn a smooth and structured latent space that encodes the complex octree structure, detailed geometry, and their relationships. Hence, our framework enables more impactful applications that are not possible with NGLOD and ACORN, such as shape completion (Fig. 8) with a partial input, shape interpolation (Fig 6), and novel shape generation (including novel octree structure, Fig. 5). We will also include more qualitative comparisons in the revision.
>
>
> ### Other Questions
> > What were the improvements that CVPR/TVCG asked, and that you applied? ...
>
> Compare with our CVPR/TVCG version, we add more evaluations (quantitative results on scene reconstruction, shape completion), and more comparisons (scene reconstruction with LIG & Conv OccNet, shape completion) to demonstrate the superiority of our method over previous works in a wider range of tasks. We will add this discussion to the supplementary.
>
> > Why did you not discuss "Neural Sparse Voxel Fields" from NeurIPS'20? While this is radiance rather than SDF based, the underlying hierarchical structure is very similar.
>
> Though NSVF also leverages sparse voxels for bounded rendering, it does not leverage hierarchies in its work -- all the octants are on the same level at each step of its progressive training. In contrast, we fully utilize hierarchical structure to deal with local shapes with different levels of geometry richness -- finer details are modeled with more local implicit functions via subdivision while coarse patches are not. In addition, we propose a novel hierarchical encoder-decoder framework to learn a generative model with meaningful latent space for shape generation and reconstruction. In comparison, NSVF is not a generative model and solely uses sparse bounded voxels for accelerated rendering. We will add this discussion to the revision.
>
> > ... strongly recommend you move the discussion of concurrent work to the intro ...
>
> We will discuss ACORD/NGLOD in the intro and make the intro/related work section clearer according to your advice.
>
> > Why is (1) more meaningful than a more classical the divergence of the gradients within the patch (which is then related to curvature)? ...
>
> We don’t claim that (1) is more meaningful than the divergence of gradients. We agree that the divergence of gradients can also achieve similar performance. We just empirically found that our current solution is simpler and can provide sufficient accuracy in our setting. We are happy to provide an ablation analysis comparing these two metrics if necessary.
>
> > Why is the VAE prior only applied to the global latent code? You have a hierarchy, how do you justify not having it at every level?
>
> We have tried adding VAE prior at each level in our early experiments. However, it turns out that the network training becomes very hard to converge as adding too many priors may sacrifice too much accuracy for regularity. However, it might be a good idea to add VAE prior to some of the intermediate levels.
>
> > Can you justify the weird failure case of LIG in Fig.4? I would expect LIG to just have slightly smoother ...
>
> To ensure a fair comparison, all the results of LIG are generated by running its officially released code. Although LIG excels at modeling large-scale 3D scenes, it struggles to capture some complex fine structures as it does not adaptively subdivide the cell according to the complexity of the enclosed geometry.
>
> > Table2: very unclear whether this is an amortized results (average) or whether it's on a single object/scene?
>
> The number of results is the average score, not a single object/scene.
>
> > ... why not compare to methods like SIREN?
>
> Though the SIREN has achieved impressive performance on shape reconstruction, it has a very different goal from our work: SIREN mainly focuses on the effectiveness of periodic activation function while our method investigates a new learnable hierarchical implicit representation for 3D shape. If requested, we can compare with SIREN using a standard implicit representation.
>
> > Sec 4.5: why are you only showing qualitative results? If no quantitative is possible, a clear rationale should be provided!
>
> We provide the quantitative results of Sec 4.5 as follows. For the table category of ShapeNet, the average CD(*$10^{-4}$) of our method and IF-Net is 4.4 and 4.9, respectively. We will add this result to the revision.

---

> > ### Comment · Reviewer_Nodw · 2021-08-23
> > **raising score**
> >
> > While I am still a bit concerned that the performance gain w.r.t. LIG is not great (~1% across metrics), I do
> > want to be more lenient in light of NGLOD (the paper will have trouble to be accepted later on).
> >
> > Regarding the ~1%, I think the best way would be to employ a synthetic dataset where a much
> > wider number of scales needs to be used to represent the geometry. In that case, LIG would
> > have to make a discrete scale decision, while in your case you could be adaptive.
> >
> > Hence I am raising my score to weak accept as I hinted in my original review.
> > I hope you will be honourable in applying the changes you mentioned in a revision!!!

---

> > > ### Author Response · Authors · 2021-08-24
> > > **Thank you!**
> > >
> > > We want to thank you for recognizing our work and for your insightful advice! We really appreciate that!!!
> > > We will compare LIG on more data where a much wider number of scales is required, as you suggested.
> > > In addition, we will apply the changes we mentioned in the revision as promised!
> > > Thank you!:-)

---

### Official Review · Reviewer_tVrM · 2021-07-12

**Rating:** 4
**Confidence:** 5

**Summary:**

The paper proposed a learnable hierarchical implicit representation for 3D learning, combined with a network architecture that can probabilistically predict hierarchical structure of 3D shape. Experiments on showed some improvements over baselines (although I have some questions, see below)

**Limitations And Societal Impact:**

There's no discussion on the social impact in the main paper

**Main Review:**

Strength:
1. The paper is tackling an important and interesting problem for 3D shape modelling as the memory consumption can increase dramatically if the resolution goes up, and the idea proposed in the paper is intersting as it  can differentially generate hierarchical structure (octree) of implicit functions, which can potentially for other tasks.
2. The paper had a wide range of experiments to demonstrate the performance including shape reconstruction, generation, interpolation, completion & scene reconstruction.
3. The paper is well-written and easy to understand

Weakness:
1. The most related work for this paper is NGLOD [48], the paper cited and discussed in the related work section (Line 109-116), however, it is very hard to claim the paper is concurrent work, as NGLOD  is already accepted to CVPR, and released almost half a year before neuritis ddl, and the paper has already released the code. The paper also argued the advantages of making the occ-tree generation to be differentiable comparing with NGLOD[48], which used a fixed oct-tree, however, in the main experiments, there's no comparisons with NGLOD that can support this argument, I'd recommend the author to add this comparison.

2. Novelty. Predicting hierarchical structure of 3D shape has been studied in many papers in computer vision literatures, as in the cited papers [49, 52, 53, 54], but they're operating on the voxels, the paper extends the ideas from these papers into the implicit function by adding an implicit decoder. Besides, considering many similar works (NeuralLoD[48], ACORN[32]), the contribution dosen't reach the high bar of neurips.

3. Minor problems: a) in table 2, using different level of both LIG and the method in the paper would have different performance, could the paper also provide the preformance of both methods at different level? otherwise it's hard to say which one is better, b) in the scene recconstruction experiments, what's the input? what exactly this experiment is doing? c) in the shape completion experiments, could the author provide quantitative comparison? looking at only two shape would have bias.

Typo: Line 314: Table 2

**Time Spent Reviewing:**

4

---

> ### Author Response · Authors · 2021-08-10
> **Response to Reviewer tVrM**
>
> We thank the reviewer for the insightful comments and suggestions! We provide our detailed response below.
>
> ### Main Questions
> > The most related work for this paper is NGLOD [48], the paper cited and discussed in the related work section (Line 109-116), however, it is very hard to claim the paper is concurrent work, as NGLOD is already accepted to CVPR, and released almost half a year before neuritis ddl, and the paper has already released the code.
>
> We claim this work is concurrent with NGLOD is because that it was concurrently submitted to CVPR 2021 (see our submission history) with NGLOD (but was unfortunately rejected due to the lack of time of polishing the writing). If the reviewer thinks it is inappropriate, we will remove this claim for better clarification.
>
> >The paper also argued the advantages of making the occ-tree generation to be differentiable comparing with NGLOD[48], which used a fixed oct-tree, however, in the main experiments, there's no comparisons with NGLOD that can support this argument, I'd recommend the author to add this comparison
>
> Making the octree generation to be differentiable allows our method to generate the most compatible octree structure with the underlying geometry, which could significantly improve the reconstruction accuracy. To verify this point, we conducuted additional experiments on scene reconstruction on 3D-Front comparing with NGLOD. The reconstruction accuracy of NGLOD is CD($\times10^{-4}$)/EMD($\times10^{-2}$)=11.3/33.1 while ours=6.4/21. The experimental results show that our method has achieved significantly higher accuracy than that of NGLOD thanks to the differentiable nature of our octree structure. We will add this quantitative comparison and visualize different octrees that we generate for varying 3D scenes for better clarification.
>
>
> > Novelty. Predicting the hierarchical structure of 3D shape has been studied in many papers in computer vision literature, as in the cited papers [49, 52, 53, 54], but they're operating on the voxels, the paper extends the ideas from these papers into the implicit function by adding an implicit decoder. Besides, considering many similar works (NeuralLoD[48], ACORN[32]), the contribution doesn’t reach the high bar of neurips.
>
> First of all, unlike NGLOD and ACORN, which only focus on reconstructing the input geometry with high accuracy, our framework is a **generative model** that is capable of 1) reconstructing input geometry with high fidelity; and 2) generating novel hierarchical octree and its corresponding geometry. With the proposed hierarchical encoder-decoder network, our method is able to learn a smooth and structured latent space that encodes the complex octree structure, detailed geometry, and their relationships. Hence, our framework enables more impactful applications that are not possible with NGLOD and ACORN, such as shape completion (Fig. 8) with a partial input, shape interpolation (Fig 6), and novel shape generation (including novel octree structure, Fig. 5).
>
> Second, different from NGLOD and ACORN, our method explicitly encodes the **entirety of the octree structure** and **the geometry features** in a **fully differentiable** manner. This cannot be trivially achieved by using any of the existing networks. We achieve this goal by proposing a novel hierarchical encoder-decoder (HED) network. Our additional experiment in the last question comparing with NGLOD has confirmed the superiority of our proposed HED network over a simple fixed encoder-decoder structure.
>
> Finally, with the proposed differential octree generative model, our framework enables some potential applications, such as **part editing** (modifying/replacing only part of the target geometry), **semantic editing** (modifying parts with the same semantic meaning or octree level), and **hierarchy editing** (change the hierarchy of the geometry by either modifying it or replacing it with a new one). Though we do not have enough time to show these applications in the paper, we have verified these are doable during rebuttal and will add these to our revision.
>
> ### Other Questions
> > In table 2, using different level of both LIG and the method in the paper would have different performance, could the paper also provide the performance of both methods at different level? otherwise it's hard to say which one is better
>
> As requested, we provide detailed comparisons on scene reconstruction with LIG at different levels below. As shown in the result, our method consistently outperforms LIG at all levels.
>
> CD($\times10^{-4}$)
>
> |level|2|3|4|5|6|
> |:-:|:-:|:-:|:-:|:-:|:-:|
> |LIG|5.21|3.27|3.23|3.15|3.02|
> |ours|4.82|2.97|2.95|2.93|2.92|
>
> > In the scene reconstruction experiments, what's the input? what exactly this experiment is doing?
>
> The input of the scene reconstruction experiment is the 3D scene geometry. In this experiment, we try to optimize the networks of different approaches and test if it can perfectly reconstruct the input after the optimization converges. For different methods, we convert the input 3D scene geometry to its compatible input representation. For our method, we convert the input into voxels.
>
> > In the shape completion experiments, could the author provide quantitative comparison? looking at only two shapes would have bias.
>
> We provide the quantitative results of Figure 8 as follows. For the table category of ShapeNet, the average CD(*$10^{-4}$) of our method and IF-Net is 4.4 and 4.9, respectively.

---

> > ### Comment · Reviewer_tVrM · 2021-08-16
> > **Tiny question for the new experiment**
> >
> > Thank the author for providing additional experiments comparing with NGLOD and new quantitative experiments.
> >
> > One more tiny question for the new experiment:
> > >The reconstruction accuracy of NGLOD is CD/EMD =11.3/33.1 while ours=6.4/21
> >
> > What the number of levels for both methods?

---

> > > ### Author Response · Authors · 2021-08-17
> > > **Answer to the new question**
> > >
> > > > What are the number of levels for both methods?
> > >
> > > The number of levels for both methods is 4, which is consistent with the experimental setting in Section 4.4.

---

> > > > ### Comment · Reviewer_tVrM · 2021-08-25
> > > > **Major concern still exists**
> > > >
> > > > Thanks for providing new answers, I agree with the author that the paper is doing a different task comparing with NGLOG, which is not generating oct-tree structure of the shape. However, this does not solve my major concerns regarding the novelty. In particular, OGN [49], which is predicting the oct-tree structure of an object, the only differences I can see comparing with OGN is 1) the paper is operating using implicit function, while OGN is using voxel, 2) the paper is predicting whether the cell is occupied and whether the cell needs to be subdivided, while the OGN is doing three categories classification: empty, filled, or mixed, 3) the paper adopted a VAE for generative power, while OGN doesn't have it. But these three differences do not novel, as implicit function and VAE are all exist techniques. I appreciate the author providing the additional comparison to OGN, but the improvement is minor (0.884 v.s. 0.898).
> > > >
> > > > > Novelty. Predicting the hierarchical structure of 3D shape has been studied in many papers in computer vision literature, as in the cited papers [49, 52, 53, 54], but they're operating on the voxels, the paper extends the ideas from these papers into the implicit function by adding an implicit decoder.
> > > >
> > > > I'm happy to listen to the author's opinion on the differences comparing with OGN, am I missing something?

---

> > > > > ### Author Response · Authors · 2021-08-26
> > > > > **Response**
> > > > >
> > > > > 1) We respectfully argue that extending voxel to implicit representation in our setting is a non-trivial task, which requires novel contributions in order to be accomplished. Since voxel is a crude approximation of the underlying shape, **OGN only needs to predict the octree itself** (i.e. the occupancy of the octree/voxel node) without the need of encoding the geometric details inside the voxel. In contrast, our task is much more challenging as **we need to encode and decode both the discrete/non-differentiable octree structure and the detailed geometry enclosed by each node in a fully differentiable manner**. This cannot be trivially achieved using any of the existing networks. We achieve this goal by proposing a novel hierarchical encoder-decoder network that ensures both the octree structure and the underlying geometric surfaces can be efficiently optimized in an end-to-end system. This enables us to encode both the entirety of the octree structure and the geometric details in a compact and meaningful latent space, which cannot be achieved by OGN.
> > > > >
> > > > > 2) Due to the limited modeling power of voxels, OGN struggles to capture tiny and intricate geometric details even at a high resolution. In comparison, thanks to the flexible and continuous nature of the implicit functions, our method can achieve 3D reconstruction and generation with much higher fidelity at an even lower cost. In addition, our method can support more useful applications that are out of the scope of OGN, such as smooth and continuous interpolation between 3D shapes, node-level geometry editing (replacing/modifying the geometry in the leaf node, or fusing the geometry of two different nodes), etc.
> > > > >
> > > > > 3) Regarding the quantitative comparison with OGN on the ShapeNet car category (0.884 v.s. 0.898), we wish to argue that this additional experiment cannot comprehensively reflect our modeling capacity. First of all, **comparing with a voxel-based method is actually unfair to our method**. To compare with OGN, we have to voxelize our reconstructed result, which is continuous and contains many fine-scale geometric details. The voxelization has significantly degraded our reconstruction quality before comparing with the voxel-based methods. Second, the experiment is conducted at the **resolution of 64** (the highest-resolution result that OGN provides in their autoencoding experiment), which is too low and has basically wiped out all of the geometric details that our method has reconstructed. We think that it would be fair to us if we are asked to compare with alternative SOTA methods based on implicit functions. And we have shown in previous experiments that our method has outperformed LIG and NGLOD significantly in the quantitative comparisons.

---

### Official Review · Reviewer_ULW2 · 2021-07-15

**Rating:** 7
**Confidence:** 3

**Summary:**

This paper describes a learned shape representation that is based on hierarchical oct trees that store local implicit functions at the voxels. The implicit functions represent the geometry. It offers higher fidelity and smaller memory footprint than using local implicits without oct trees (LIG, [26]). In comparison to other oct-tree based methods, the proposed approach offers the ability to learn oct-tree hierarchy.

**Limitations And Societal Impact:**

Acceptable

**Main Review:**

The proposed method is technically sound and achieves good results and improvements over the state of the art. Ability to learn (encode and decode) the hierarchical structure of the octtree, seems like an interesting contribution that can be used in other spatial representations, beyond storing implicit functions. Quantitatively, the proposed method also offers improvements over previous work.

I suggest authors revise their related work section to discuss prior learning techniques that used OctTrees in more details. Right now  only OpenVDB [39] and NGLOD [48] are discussed, while lots of highly relevant techniques [52, 54, 53, 49] are only casually mentioned on line 91. I think they need to be discussed more, especially [54] which is further used as a baseline.  Paper "Hierarchical surface prediction for 3d object reconstruction" [24] is only mentioned in the context of data pre-processing, but seems highly related.

**Time Spent Reviewing:**

1

---

> ### Author Response · Authors · 2021-08-10
> **Response to Reviewer ULW2**
>
> We thank the reviewer for the insightful comments and the positive feedback! We provide our detailed response below.
>
> > I suggest authors revise their related work section to discuss prior learning techniques that used OctTrees in more details. Right now only OpenVDB [39] and NGLOD [48] are discussed, while lots of highly relevant techniques [52, 54, 53, 49] are only casually mentioned on line 91. I think they need to be discussed more, especially [54] which is further used as a baseline. Paper "Hierarchical surface prediction for 3d object reconstruction" [24] is only mentioned in the context of data pre-processing, but seems highly related.
>
> We will discuss them in detail in the revision.
> OGN[49] aims to generate high-resolution 3D voxels by octrees in a multi-scale fashion, which is also computation- and memory-efficient. The pioneering work O-CNN [52] first adopts the octree to the shape analysis (classification, retrieval, part segmentation), which represents the shape as octree of voxels and develops the specially-tailored convolution and pooling layers to accommodate the complex structure of octree.  Based on the O-CNN, Adaptive-OCNN [54] makes the octree subdivision adaptive for the richness of the geometry details. For each octant, it uses a planar patch to fit the local surface. To improve the modeling efficiency for large-scale data, [53] extends them with skip-connection to a very deep O-CNN network, which achieves significant improvements in prediction accuracy using voxel representation. In comparison, we propose a novel hierarchical encoder-decoder network that is specially tailored for implicit function and is able to encode the entirety of the octree structure along with the geometry features in a fully differentiable manner.
> [24] takes the image, depth, or partial volume as inputs and generates the high-resolution voxels from coarse to fine in an octree hierarchically. However, [24] focuses on predicting fine-grained voxels while our approach aims to generate hierarchical implicit fields at each octant which can represent fine geometry with lower cost.

---

### Official Review · Reviewer_2BPG · 2021-07-16

**Rating:** 6
**Confidence:** 5

**Summary:**

This paper presents a new efficient 3D shape representation to reduce computation cost while also being able to preserve geometric details. The authors propose to represent 3D shapes using octree structures such that the hierarchies focus on modeling the surface geometry, where the leaf nodes additionally predicts implicit functions that returns the occupancy for each query 3D point. Both the hierarchical inference and prediction are designed to be differentiable such that the octree structures can be discovered and learned from data. Experiments show that the proposed representation achieves the state of the art on the ShapeNet reconstruction with applications of shape interpolation and completion.

**Limitations And Societal Impact:**

Adequately addressed

**Main Review:**

Strengths:
+ This work combines a neat idea of blending octrees (often used for efficient voxelized shape inference/prediction) and implicit surfaces (for modeling dense and granular surface structures). It saves unnecessary computational costs by terminating early in monotonic occupancy/freespace regions to focus more on modeling the surfaces. I like how the hierarchy can be handled with MLPs, which would be much easier to parallelize while being adaptive to different numbers of input nodes, unlike sparse 3D convolution which requires specialized designs.
+ The use of additional variables $\boldsymbol{\alpha}$ and $\boldsymbol{\beta}$ for predicting recursive octree structures is nice, whose differentiability allows end-to-end training.
+ The proposed method achieves the state of the art performance on the ShapeNet reconstruction benchmark, and the new representation shows good results on scene reconstruction of indoor room scales as well.

Weaknesses:
- It is unclear why $\boldsymbol{\alpha}$ and $\boldsymbol{\beta}$ are necessary in the encoder; maintaining architectural symmetry does not seem important here. Furthermore, it is not clear why the hierarchical encoder is important. One can preprocess the input shapes into different 3D representations (meshes, point clouds, naive voxels) and use other encoders to get the latent codes, and still use the hierarchical decoder for shape reconstruction. The effect of the proposed encoder architecture is not very related to 3D shape predictions as presented in the experiments. To validate that the new encoder indeed learns a better latent space, the authors should run another experiment on comparing the learned latent space, e.g. by fitting a linear classifier on the trained latent codes as a measure of discrimination and generalization.
- The VAE reparametrization seems like an orthogonal component to the paper and only allows the application random shape generation. I would suggest removing it, as the experiments are more focused on shape reconstruction.
- Are the implicit occupancy decoders (at the leaf nodes) shared? I am assuming yes (like the encoders), but it is not mentioned anywhere in the paper. Also, are the intermediate MLPs shared as well? If so, would it be possible to apply the MLPs (in the decoder part) for an arbitrary number of times until the desired level of details is reached, adaptive to the threshold $\tau$?
- An ablation on the $\tau$ value should be included to better understand its effectiveness. The expectation is that when $\tau=1$, the proposed decoder would become OccNet [34]; if $\tau=0$, it would become OGN [49].
- In Sec 4.1, was a single network trained on multiple shape categories together, or was it category-specific?
- I'm confused by L280-282. Why was SDF computed where (binary) occupancy prediction is the target goal?
- It would be helpful to also compare the ShapeNet reconstruction against OGN [49], as it shares a very similar octree generating architecture but for voxels.
- The setting in the scene reconstruction experiment is unclear. A brief overview of the 3D-Front dataset should be provided (e.g. how many train/test examples). It would also be helpful to provide quantitative results and more visual comparisons for this experiment.

Other minor problems:
- It would be helpful to enlarge Fig. 4, as it is difficult to compare in detail. A page-width figure would be recommended. It would also be helpful to visualize important parts where the proposed method is significantly superior to the baseline methods (e.g. the bars of the chairs).
- L129: DeepSDF should not be considered as concurrent work, which is a bit misleading. It would also be helpful to elaborate how this work is different from ACORN [32], as both works share a large similarity in hierarchical modeling of signals (e.g. 3D shapes).
- L28-29: I don't think voxels are explicit representations since it encodes occupancy at 3D locations, which is different from meshes or point clouds.
- L247: the notations $\boldsymbol{\alpha}$ and $\boldsymbol{\beta}$ are overloaded with that from the encoder. It is suggested to use a slightly different notation to discriminate, e.g. $\boldsymbol{\hat{\alpha}}$.
- An important reference is missing: OctNet [A], which was the first to use an octree structure for inferring 3D shapes.

[A] Riegler et al. "OctNet: Learning Deep 3D Representations at High Resolutions." CVPR 2017

**Time Spent Reviewing:**

5hr

---

> ### Author Response · Authors · 2021-08-10
> **Response to Reviewer 2BPG**
>
> We thank the reviewer for the insightful comments and suggestions! We provide our detailed response below.
>
> ### Main Questions
> > It is unclear why $\alpha$ and $\beta$ are necessary in the encoder; maintaining architectural symmetry does not seem important here. Furthermore, it is not clear why the hierarchical encoder is important. ... run another experiment on comparing the learned latent space, e.g. by fitting a linear classifier on the trained latent codes as a measure of discrimination and generalization.
>
> We agree that our framework is compatible with other encoders without architectural symmetry. However, by incorporating $\alpha$ and $\beta$ in a hierarchical manner, which explicitly encodes the structural information and the hierarchy of the octree and the geometry, it leads to more discriminative features in the latent space, that can significantly boost the performance of the subsequent applications, e.g shape reconstruction. We verify this point by conducting a control experiment that replaces the proposed hierarchical encoder with a PointNet++ encoder for encoding the point cloud. On the task of shape reconstruction, experimental results show that our encoder has significantly outperformed Pointnet++ in terms of Chamfer distance ($\times 10^{-4}$) on the Shapenet chair category: **our hierarchical encoder: 2.19 vs. Pointnet++ encoder: 13.54**. Regarding the task of shape generation, the generative ability of the Pointnet++ based encoder is much worse than our proposed method (e.g. fails to generate diversified shapes via random sampling). Both experiments have demonstrated that without incorporating the structural hierarchies and the architectural symmetry in the encoder, it is difficult to decode the complex structure of octree and the geometric details. All the results (quantitative and qualitative) will be reported in the revised version.
>
> For the experiment on applying a linear classifier on the learned latent space of our encoder, we conduct a binary classification experiment on two large categories: chair and table.
> Though our network is not tailored for classification, our approach still achieved a classification accuracy of 93.2%. The t-SNE visualization also shows that our method can generate discriminative features even with the high geometry similarity between chairs and tables.
>
> These two experiments are interesting and important for our novel contribution, we will add full experiments in our revised version.
>
> > The VAE reparametrization seems like an orthogonal component to the paper and only allows the application random shape generation. I would suggest removing it, as the experiments are more focused on shape reconstruction.
>
> Thanks for the suggestion! However, in addition to shape reconstruction, learning a generative model that can produce diverse geometries with fine-grained details is also the main objective of our work. Hence, we encourage the network to learn a smooth and more structured latent space for shape generation and interpolation (see Figure 5 and 6). In this sense, VAE is an essential component.
>
>
> > Are the implicit occupancy decoders (at the leaf nodes) shared? ... Also, are the intermediate MLPs shared as well? If so, would it be possible to apply the MLPs (in the decoder part) for an arbitrary number of times until the desired level of details is reached, adaptive to the threshold $\tau$?
>
> Both the implicit occupancy decoders (at the leaf nodes) and the intermediate MLPs are shared. We will clarify them in the revised paper.
> Yes, our framework can support applying MLPs for an arbitrary number of times to achieve the desired level of details by controlling the octree depth, which is adaptive to the threshold $\tau$.
>
> > An ablation on the $\tau$ value should be included ... The expectation is that when $\tau=1$, the proposed decoder would become OccNet [34]; if $\tau=0$, it would become OGN [49].
>
> The division depends on not only the value of $\tau$, but also a maximum depth. The main purpose of introducing $\tau$ is to reduce the division without significantly affecting the accuracy. If the $\tau$ value is set to 1, then the root node will not be subdivided and our method will indeed become equivalent with an OccNet. However, if the $\tau$ value is set to 0, the octree will be divided to the maximum depth. The leaf node still uses an implicit function to represent the shape, which is different from OGN that uses voxels. Furthermore, the ablation on $\tau$ is listed below and will be added to our revision.
>
> |$\tau$|0|0.1|0.5|0.8|
> |:-:|:-:|:-:|:-:|:-:|
> |CD($\times10^{-4}$)|2.96|2.97|4.87|8.42|
>
> > ..., was a single network trained on multiple shape categories together, or was it category-specific?
>
> We train our network in a category-specific manner.
>
> > ...Why was SDF computed where (binary) occupancy prediction is the target goal?
>
> In the pre-processing, we use the sign of calculated SDF as the input and predict the occupancy in the decoder.
>
> > It would be helpful to also compare the ShapeNet reconstruction against OGN [49], ...
>
> We tested our method on the ShapeNet-Car dataset used by OGN. The comparison of the IoU accuracy at $64^3$ resolution with the value reported in the OGN paper is as follows: OGN-0.884 v.s. ours-0.898 (the higher the better). We will include this comparison in the revision.
>
> > The setting in the scene reconstruction experiment is unclear. A brief overview of the 3D-Front dataset should be provided ... It would also be helpful to provide quantitative results and more visual comparisons for this experiment.
>
> The 3D-Front is a newly released dataset of 3D indoor scenes which contains 6815 houses and 51708 rooms. The furnished objects come from 3D-Future, a dataset of textured 3D furniture models. The dataset is divided into the training and testing sets with a ratio of 3:1. We have provided the quantitative results in Table 3 in our paper, where we compare with LIG and ConvOCCNet on scene reconstruction. We will add more visual comparisons to the revised paper/supp.
>
>
> ### Other Questions
> > It would be helpful to enlarge Fig. 4, ...
>
> We will enlarge our figures or put them into our supplementary for a clearer presentation.
>
> > L129: DeepSDF should not be considered as concurrent work, which is a bit misleading. It would also be helpful to elaborate how this work is different from ACORN, ....
>
> We were intending to mean that DeepSDF is a concurrent work with IM-Net in the context. We will remove this statement to avoid confusion.
>
> Our framework is a generative model that can produce a novel hierarchical octree and its corresponding geometry. Our network learns a smooth and structured latent space that encodes the complex octree structure, detailed geometry, and their relationships. Hence, our method is able to support many useful applications, such as shape completion (Fig. 8) with given partial input, shape interpolation (Fig. 6), novel shape generation (including novel octree structure), and shape editing. In contrast, ACORN only focuses on shape reconstruction with higher accuracy and fine geometric details.
>
>
> > L28-29: I don't think voxels are explicit representations since it encodes occupancy at 3D locations, ...
>
> Voxel is indeed an explicit representation in traditional definition as it explicitly casts a 3D shape with stacked voxels, indicating they belong to the modeled object or its separate parts. This is also agreed by the latest paper [1] (see the abstract) that utilizes voxel representation. However, if the voxel positions are viewed as sampling positions in an implicit field, it can indeed be considered an implicit representation.
>
> [1] Neural Sparse Voxel Fields, NeurIPS 2020
>
> > L247: the notations $\alpha$ and $\beta$ are overloaded with that from the encoder. ...
>
> Thank you for the suggestion! We will refine our equation formulation according to your advice.
>
> > An important reference is missing: OctNet, ...
>
> We will cite and discuss it.

---

### Decision · Program_Chairs · 2021-09-27

**Decision:**

Accept (Poster)

**Comment:**

Post rebuttal, the paper was the subject of extensive discussion both between the authors and reviewers and between the reviewers themselves. The reviewers were overall generally in agreement with many facts about the paper, but had good faith disagreements in terms of where to draw boundaries for novelty and contribution. The AC examined the reviews, rebuttals, and discussion and is inclined to agree with the reviewers advocating for acceptance. The AC is persuaded by overall perspective of reviewers 2BPG and Nodw that the paper contributes valuable insights to the field (presented in committee discussions). Moreover, if the reviewers are satisfied that their clarity concerns can be addressed in a revision, then the AC is as well.

The AC would make the following requests:
- The authors should carefully read the comments of the reviewers and be sure each commented is addressed in the final version of the paper. The rebuttal and a promise of revision was persuasive in this case.
- The authors should add a stronger discussion of the social impact in the paper.